# Breaking Free from MMI: A New Frontier in Rationalization by Probing Input Utilization

**Wei Liu**[1], **Zhiying Deng**[2][*], **Zhongyu Niu**[1], **Jun Wang**[3][*], **Haozhao Wang**[1][*],
**Zhigang Zeng**[4], **Ruixuan Li**[1][*]

[1]School of Computer Science and Technology, HUST

[2] Faculty of Artificial Intelligence in Education, Central China Normal University

[3]iWudao Tech    [4]School of Artificial Intelligence and Automation, HUST

[1]{idc_lw,zy_niu, hz_wang, rxli}@hust.edu.cn

[2]zhiyingdzy@gmail.com    [3]jwang@iwudao.tech,    [4]zgzeng@hust.edu.cn

## Abstract

Extracting a small subset of crucial rationales from the full input is a key problem in explainability research. The most widely used fundamental criterion for rationale extraction is the maximum mutual information (MMI) criterion. In this paper, we first demonstrate that MMI suffers from diminishing marginal returns. Once part of the rationale has been identified, finding the remaining portions contributes only marginally to increasing the mutual information, making it difficult to use MMI to locate the rest. In contrast to MMI that aims to reproduce the prediction, we seek to identify the parts of the input that the network can actually utilize. This is achieved by comparing how different rationale candidates match the capability space of the weight matrix. The weight matrix of a neural network is typically low-rank, meaning that the linear combinations of its column vectors can only cover part of the directions in a high-dimensional space (high-dimension: the dimensions of an input vector). If an input is fully utilized by the network, it generally matches these directions (e.g., a portion of a hypersphere), resulting in a representation with a high norm. Conversely, if an input primarily falls outside (orthogonal to) these directions, its representation norm will approach zero, behaving like noise that the network cannot effectively utilize. Building on this, we propose using the norms of rationale candidates as an alternative objective to MMI. Through experiments on four text classification datasets and one graph classification dataset using three network architectures (GRUs, BERT, and GCN), we show that our method outperforms MMI and its improved variants in identifying better rationales. We also compare our method with a representative LLM (llama-3.1-8b-instruct) and find that our simple method gets comparable results to it and can sometimes even outperform it. Code: https://github.com/jugechengzi/Rationalization-N2R.

## 1 Introduction

With the success of deep learning, there are growing concerns over the model interpretability. Exploring the theory and technique of interpretable machine learning frameworks is of immense importance in addressing a myriad of issues. For instance, XAI techniques can aid in detecting model discrimination (fairness) (Pradhan et al., 2022), identifying backdoor attacks (security) (Li et al., 2022), and revealing potential failure cases (robustness) (Chen et al., 2022; Zhang et al., 2024), among others. Post-hoc explanations, which are trained separately from the prediction process, may not faithfully represent an agent's decision, despite appearing plausible (Lipton, 2018). In contrast to post-hoc methods, ante-hoc (or self-explaining) techniques typically offer increased transparency (Lipton, 2018) and faithfulness (Yu et al., 2021), as the prediction is made based on the explanation itself. There is a stream of research that has exposed the unreliability of post-hoc explanations and called for self-explanatory methods (Rudin, 2019; Ghassemi et al., 2021; Ren et al., 2024).

---

[*]Corresponding authors.

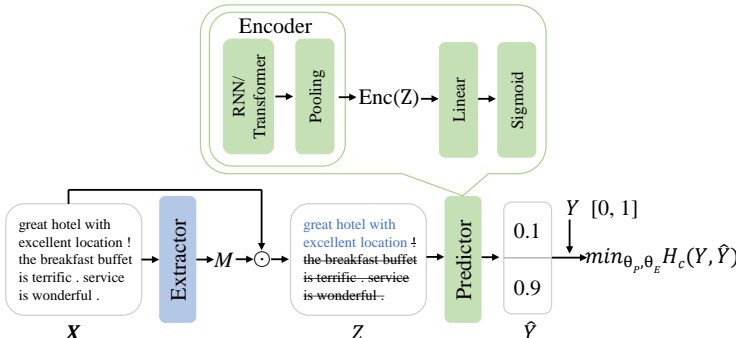

Figure 1: The standard rationalization framework RNP. The task is binary sentiment classification about the hotel's location. $X, Z, \hat{Y}, Y$ represent the input, the extracted rationale candidate, the prediction and the ground truth label, respectively. $M$ is a sequence of binary masks. $Enc(Z)$ is the encoder's final layer representation (like the term "embedding" in (Lee et al., 2024; Bolukbasi et al., 2021)). $\theta_E, \theta_P$ represent the parameters of the extractor and the predictor. $H_c$ denotes cross-entropy.

In this study, our primary focus is on investigating a general model-agnostic self-explaining framework called Rationalizing Neural Predictions (RNP, also known as rationalization) (Lei et al., 2016), which with its variants has become one of the mainstream methods to facilitate the interpretability of NLP models (Sha et al., 2021; Yu et al., 2021; Antognini et al., 2021; Liu et al., 2022; 2023a;c;b; 2024b;a; Storek et al., 2023; Liu et al., 2024c; Zhao et al., 2024), and also holds the potential to be applied to image classification (Yuan et al., 2022) and graph neural networks (Luo et al., 2020). RNP utilizes a cooperative game involving an extractor and a predictor. This game is designed with a focus on "data-centric" (i.e., it is to explain the connection between a text and the model-agnostic task label, rather than explaining the output of a specific model) feature importance. The extractor first identifies the most informative part of the input, termed the rationale. Subsequently, the rationale is transmitted to the predictor to make predictions, as illustrated in Figure 1. Apart from its use for interpretability, some recent studies find that rationalization can also serve as a method for data cleaning. The extracted $(Z, Y)$ pairs can act as a new dataset, and trained with such a cleaned dataset, a predictor may be more robust (Chen et al., 2022) and generalizable (Wu et al., 2022; Gui et al., 2023), thanks to the removal of task-irrelevant, harmful information.

The commonly used objective for finding rationales is maximizing the mutual information between the rationale candidates and the task labels (e.g., by minimizing cross-entropy), which is called the maximum mutual information (MMI) criterion. In practice, however, MMI faces the problem of diminishing marginal utility. For example, if the rationale candidate consists of $80\%$ the real rationales and $20\%$ random noise, it might be informative enough to help a predictor make the correct prediction (see a specific toy example in §4.1, and empirical verification in Figure 3(a)(b)). In this case, replacing the left $20\%$ noise with real rationales can only trivially improve the mutual information (or intuitively, the prediction accuracy. See §4.1 for the theoretical perspective). As a result, the gradient provided by MMI cannot guide to find the left $20\%$ real rationales well.

To avoid the shortcomings of MMI, instead of following the traditional approaches of fixing MMI's flaws through various regularization terms, this paper aims to find an alternative objective to the MMI criterion. Since the MMI criterion has been the fundamental objective in the XAI literature for a long time, our finding is important as it opens a new avenue for extractive interpretability without the fundamental MMI. The overall idea can be summarized as follows: (1) The rationales are those the predictor can learn and utilize. (2) Neural networks usually have low-rank weight matrices (Kang et al., 2024), meaning that the linear combinations of its column vectors can only cover part of the directions in a high-dimensional space (high-dimension: the dimensions of an input vector). (3) For features that the network does not learn, their directions are usually orthogonal to the learned directions (see Appendix A.14), and the representation norms through the weight matrix will approach zero. On the contrary, well-learned features usually interact with the learned directions of the weight matrix, resulting in representations with higher norms (Appendix A.15 provides a toy example for intuitive understanding). This property is borrowed from recent advances (Kang et al., 2024) in the out-of-distribution (OOD) detection field. Based on the above properties, we attempt

to find rationales by maximizing the norm (i.e., $\|Enc(Z)\|_2$ in Figure 1) of the rationale candidate's representation, whose motivation is further empirically supported by the results in Figure 3(c).

The key difference between our method and the mainstream MMI approaches can be summarized at a high level: MMI aims to find rationales that can reproduce the prediction, while our study seeks to identify the parts of the input that the network can actually utilize (i.e., match the non-zero rank subspaces of the weight matrix), whose idea is more in line with the philosophy of **ante-hoc** explanation. Our work opens new eyes in the XAI literature, as almost all current mainstream XAI methods follow the MMI criterion or its variants. The idea to observe which parts of the input can be utilized by the network from the perspective of forward propagation is also novel, as this is the first time that an extractive XAI method has been freed from relying on the model's final output (e.g., may be used in the future to probe task-irrelevant (i.e., not fine-tuned) pretrained encoders. But this is beyond the scope of this paper and is left for future work.).

The contributions can be summarized as follows: (1) We empirically find the diminishing marginal utility problem of identifying rationales with the MMI criterion (Figure 3). Considering that MMI is a widely used criterion for finding explanations, this empirical observation may somewhat remind the XAI community to rethink this fundamental criterion. (2) We formally analyze the reasons why the diminishing marginal utility can occur with the MMI criterion, providing insights for further researchers to better address this issue. (3) Based on a theoretical property borrowed from the OOD research, we propose an alternative objective to the MMI criterion. Empirical results on both text and graph data with three different encoders (GRUs, BERT, and GCN) show that our method not only outperforms the vanilla MMI, but also beats its several recently improved variants.

## 2 RELATED WORK

**Extractive Rationalization**. The self-explaining framework of rationalization named RNP (Lei et al., 2016) is flexible and offers a unique advantage: certification of exclusion, which means any unselected input is guaranteed to have no contribution to prediction, making it important to the NLP community (Yu et al., 2021). Based on it, many methods have been proposed to improve RNP from different aspects. Bao et al. (2018) used Gumbel-softmax to do the reparameterization for binarized selection. Bastings et al. (2019) replaced the Bernoulli sampling distributions with rectified Kumaraswamy distributions. Jain et al. (2020) disconnected the training regimes of the generator and predictor networks using a saliency threshold. Paranjape et al. (2020) imposed a discrete bottleneck objective to balance the task performance and the rationale length. DeYoung et al. (2020) proposed a benchmark that can be used for supervised rationale extraction. 3PLAYER (Yu et al., 2019) tries to squeeze the informative texts from the unselected parts to produce comprehensive rationales. DMR (Huang et al., 2021) tries to align the distributions of rationale with the full input text in both the output space and feature space. A2R (Yu et al., 2021) endows the predictor with the information of full text by introducing a soft rationale. FR (Liu et al., 2022) folds the two players to regularize the predictor with the extractor (as the extractor can view the raw input) by sharing a unified encoder. Inter_RAT (Yue et al., 2023) tried to use backdoor adjustment to alleviate the spurious correlations in the raw dataset. Fernandes et al. (2022) leveraged meta-learning techniques to improve the quality of the explanations. Havrylov et al. (2019) cooperatively trained the models with continuous and discrete optimisation schemes. (Hase et al., 2020) explored better metrics for evaluation. (Rajagopal et al., 2021) used phrase-based concepts to conduct a self-explaining model. Other methods like data augmentation with pretrained models (Plyler et al., 2021), training with human-annotated rationales (Chan et al., 2022), injecting noise to the selected rationales (Storek et al., 2023), using attack techniques to inspect spurious correlations (Liu et al., 2024c), have also been tried.

All of these previous studies take MMI as the fundamental criterion of finding rationales, and the diminishing marginal utility problem has been overlooked. The purpose of this paper is to analyze the diminishing marginal utility problem and to alleviate it, which is orthogonal to previous research.

**The properties of the network's learned inputs**. Some previous research has found that complex neural networks typically have low-rank weight matrices (Aghajanyan et al., 2021). Kang et al. (2024) shows both theoretically and empirically that the weight matrices and network representations associated with the learned inputs often occupy low-dimensional subspaces with high overlap. However, when the network encounters unlearned OOD inputs, their associated representations tend to have less overlap with the weight matrices compared to those the network has learned. As a re-

sult, the feature representations corresponding to the unlearned OOD inputs tend to have smaller norms than those of the learned inputs, resulting in less signal being propagated from the input. Our method is inspired by this theoretical property.

**Generative explanation with LLMs**. Generative explanation is a research line that is close but orthogonal to our research on extractive explanation. With the great success of LLMs, a new research line for explanation is chain-of-thought. By generating (in contrast to selecting) intermediate reasoning steps before inferring the answer, the reasoning steps can be seen as a kind of explanation. The intriguing technique is called chain-of-thought (CoT) reasoning (Wei et al., 2022). However, LLMs sometimes exhibit unpredictable failure modes (Kıcıman et al., 2023) or hallucination reasoning (Ji et al., 2023), making this kind of generative explanation not trustworthy enough in some high-stakes scenarios. Also, some recent research finds that LLMs are not good at extractive tasks (Qin et al., 2023; Li et al., 2023; Ye et al., 2023).

**The potential impact of rationalization in the era of LLMs**. Compared to traditional "model-centric" XAI methods which solely focus on the model's learned information, "data-centric" approaches primarily aim to extract model-agnostic patterns inherent in the data. So, apart from improving interpretability, rationalization can serve as a method of data cleaning Seiler (2023).

Domain-specific large models often require supervised fine-tuning using domain-specific data. Uncleaned data may contain harmful information such as biases and stereotypes (Sun et al., 2024). Recent research suggests that training predictors with extracted rationales can remove irrelevant harmful information, enhancing robustness (Chen et al., 2022) and generalization (Wu et al., 2022; Gui et al., 2023). Considering that small models are sufficient for simple supervised tasks and are more flexible and cost-effective for training on single datasets (e.g., searching hyperparameters and adding auxiliary regularizers), using small models for rationalization on a single dataset and then using the extracted rationales for supervised fine-tuning might prevent large models from learning harmful information from new data. Additionally, shortening input texts can also reduce the memory required for fine-tuning (Guan et al., 2022). A recent study also finds that training a small model for data selection (although not the same as rationale selection) and producing a small subset is useful for fine-tuning LLMs (Xia et al., 2024).

We compare our method against a representative LLM (llama-3.1-8b-instruct), in Appendix A.7, and demonstrate that our approach achieves comparable results, sometimes even surpassing it.

## 3 PRELIMINARIES

### 3.1 THE RATIONALE EXTRACTION TASK

We consider the text classification task, where the input is a text sequence $X=[x_1, x_2, \cdots, x_l]$ with $x_i$ being the $i$-th token and $l$ being the number of tokens. $Y$ represents the classes in a dataset $\mathcal{D}$. The standard rationalization framework RNP (Lei et al., 2016) consists of an extractor $f_E(\cdot)$ and a predictor $f_P(\cdot)$, with $\theta_e$ and $\theta_p$ representing the parameters of the extractor and predictor. For $(X, Y) \sim \mathcal{D}$, the extractor first outputs a sequence of binary mask $M = f_E(X) = [m_1, \cdots, m_l] \in \{0,1\}^l$ (in practice, the extractor first outputs a Bernoulli distribution for each token and the mask for each token is independently sampled using gumbel-softmax). Then, it forms the rationale candidate $Z$ by the element-wise product of $X$ and $M$:

$$Z = M \odot X = [m_1 x_1, \cdots, m_l x_l]. \tag{1}$$

To simplify the notation, we denote $f_E(X)$ as $Z$ in the following sections, i.e., $f_E(X) = Z$. With the extractor's selection, we get a set of $(Z, Y)$ samples, which are generally considered to represent the distribution $P(Y|Z)$. The rationale $Z$ is searched by maximizing the mutual information $I(Y;Z)$:

$$Z^* = \arg\max_Z I(Y;Z) = \arg\max_Z (H(Y) - H(Y|Z)) = \arg\min_Z H(Y|Z), \ s.t., \ Z = f_E(X).$$
$$\tag{2}$$

In practice, the entropy $H(Y|Z)$ is commonly approximated by the minimum cross-entropy $\min_{\theta_p} H_c(Y, \hat{Y}|Z)$, with $\hat{Y} = f_P(Z)$ representing the output of the predictor. It is essential to note that the minimum cross-entropy is equal to the entropy (please refer to Appendix A.5).

As a result, the predictor uses the cross-entropy objective to do the classification, and the extractor also uses the cross-entropy objective to find good rationales:

$$\textbf{Extractor:} \min_{\theta_e} H_c(Y, f_P(Z)|Z)$$

$$\textbf{Predictor:} \min_{\theta_p} H_c(Y, f_P(Z)|Z) \tag{3}$$

$$s.t., \ Z = f_E(X), \ (X, Y) \sim \mathcal{D}.$$

Replacing $Z$ with $f_E(X)$, the extractor and the predictor are trained cooperatively to minimize the cross-entropy. Here we rewrite Equation (3) for better conciseness and clarity:

$$\min_{\theta_e, \theta_p} H_c(Y, f_P(f_E(X))|f_E(X)), \ s.t., \ (X, Y) \sim \mathcal{D}. \tag{4}$$

To make the selected rationale human-intelligible, rationalization methods usually constrain the rationales by compact and coherent regularization terms. In this paper, we use the most widely used constraints proposed by Chang et al. (2020):

$$\Omega(M) = \lambda_1 \left| \frac{\|M\|_1}{l} - s \right| + \lambda_2 \sum_{t=2}^{l} |m_t - m_{t-1}|. \tag{5}$$

The first term encourages that the percentage of the tokens being selected as rationales is close to a pre-defined level $s$. The second term encourages the rationales to be coherent.

## 3.2 THE PROPERTIES OF THE NETWORK'S UTILIZATION ON DIFFERENT INPUTS

It has been found that neural networks usually have low-rank weight matrices (Kang et al., 2024). For these low-rank weight matrices, they correspond to low-dimensional subspaces. And the location and magnitude of these subspaces are determined by the information they learn (Kang et al., 2024). If a network just learns (or is trained on) some uninformative noise, the rank will be low and the corresponding subspace will be narrow. On the contrary, if the network learns from informative knowledge, the subspace will be wider. From a localization perspective, the learned inputs often occupy these low-dimensional subspaces with high overlap. And the unlearned inputs tend to have little overlap. As a result, if an input is informative and the knowledge is learned by the network, the representation of it through the network will have a high $l_2$ norm. If an input does not contain the knowledge learned by the network, the norm will approach $0$ (Kang et al., 2024).

Simply put, the rank of the weight matrix is determined by the knowledge learned by the network and is reflected in the directions on the hypersphere that can be occupied by the combination of the column vectors (including position and size). These areas occupy a subspace in the high-dimensional space, which we refer to as the capability subspace of the weight matrix. If an input contains features that the network has learned, it is highly likely to fall within the capability subspace and match the learned directions, leading to a representation with a higher norm. On the other hand, if an input does not contain the learned information, it is likely to fall outside the capability subspace and be orthogonal to the column vectors (see Appendix A.14), thus its representation norm will be very low, behaving like noise. At the same time, if the network does not learn any informative knowledge, the capability subspace of the weight matrix itself will be very low and all inputs will have low norms.

This property was first used by Kang et al. (2024) to remove out-of-distribution inputs in reinforcement learning. But we think it can also be used for interpretability. By observing the norms of the representations of different rationale candidates, we can determine how well they match the network, and thus identify which parts of the full input the network is actually utilizing.

## 4 THE LIMITATIONS OF MINIMIZING CROSS-ENTROPY AND OUR METHOD

### 4.1 THE DIMINISHING MARGINAL RETURNS

Although using MMI to identify rationales has almost become the default choice, we find that it faces the problem of diminishing marginal returns. Once the majority of the rationale components have

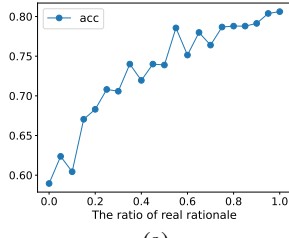 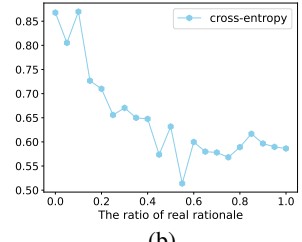 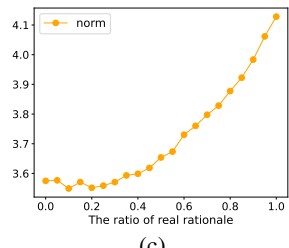

(a)       (b)       (c)

Figure 3: The (a) prediction accuracy, (b) cross-entropy loss, and (c) the norm of the representation (i.e., $\|Enc(Z)\|_2$) through the neural network vary with the proportion of true rationale components in the rationale candidate input within a trained standard RNP predictor. The dataset is *Beer-Aroma*. The results of more datasets are shown in Appendix A.8.

been identified, discovering the left rationale components has a minimal effect on further reducing cross-entropy.

In this section, we first provide an intuitive toy example to help readers understand the diminishing marginal returns from a high-level intuition. Then, we show the origin of the diminishing marginal returns problem. Finally, we provide empirical evidence from real-world datasets to verify the existence of this problem in practice.

**An intuitive toy example**. Consider a comment about food "······ The food is very delicious, and I like it very much. ······" and we need to predict its sentiment label. Both $R_1$ ="The food is very delicious" and $R_2$ ="and I like it very much" can indicate a positive enough sentiment tendency. However, if either $R_1$ or $R_2$ is given, finding another component will not contribute much to the sentiment polarity . This creates an obstacle to finding the complete rationale by MMI (minimizing the cross-entropy in practice). A more visual example is provided in Figure 2. The distance between $R_1$ and $R_1\&R_2$ is non-trivial, so the extractor needs to pay considerable effort to move from the state of selecting $R_1$ to the state of selecting both $R_1$ and $R_2$. However, the payoff for this effort is minimal, resulting in small gradients provided by the gradient descent algorithm for this move.

The reasons of this problem can lie in two aspects. One is the gradient saturation problem of the sigmoid function before the predictor's output $\hat{Y}$, which is quite intuitive and can be illustrated by the example in Figure 2 (softmax is similar). Aside from the sigmoid function, the problem can also be introduced by the diminishing marginal returns of mutual information itself.

**The theoretical perspective**. The high-level understanding of mutual information $I(Y;Z)$ is that, how much the uncertainty of $Y$ decreases when given $Z$.

We consider a perfect rationale $R$ composed of $R_1$ and $R_2$. The problem is that MMI does not satisfy additivity, which means that we cannot promise $I(Y;R_1,R_2)$ = $I(Y;R_1) + I(Y;R_2)$. The combined effect of $R_1$ and $R_2$ on reducing the uncertainty of $Y$ may be less than the sum of the individual effects of $R_1$ and $R_2$ on reducing the uncertainty of $Y$. Formally, we consider the situations where

$$I(Y;R_1,R_2) \le I(Y;R_1) + I(Y;R_2). \quad (6)$$

Please refer to Appendix A.6 for more detailed discussions about Equation (6).

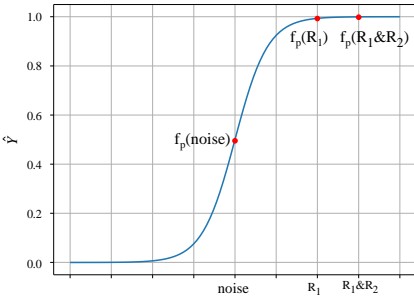

Figure 2: The diminishing marginal returns in Sigmoid function.

From Equation (6), we can further get

$$
\begin{aligned}
I(Y;R_2|R_1) &= I(Y;R_2,R_1) - I(Y;R_1) \\
&\le [I(Y;R_1) + I(Y;R_2)] - I(Y;R_1) \\
&= I(Y;R_2).
\end{aligned}
\quad (7)
$$

It means that, although $R_2$ is informative and can reduce the uncertainty of $Y$, it can be less effective if conditioned on another informative part $R_1$.

**Empirical verification of the diminishing marginal returns**. Figure 3 shows how the prediction accuracy, cross-entropy loss, and the norm of the representation (i.e., $\|Enc(Z)\|_2$ in Figure 1) vary with the proportion of true rationale components in the rationale candidate input within a trained standard RNP predictor. The dataset is a text classification dataset *Beer-Aroma*. The results of more datasets are shown in Appendix A.8.

The test set contains human-annotated ground-truth rationales. We base our inputs on the manually annotated ground-truth gold rationales in the original full text. A certain proportion of tokens in the gold rationales are replaced with tokens randomly selected from the non-rationale part of the same full text. The x-axis represents the proportion of tokens that have not been replaced, where 0 indicates all tokens are random, and 1 indicates the complete ground-truth rationale. It can be observed that when the rationale candidate used as input contains more than $60\%$ true rationale components, the decrease in cross-entropy loss (Figure 3(b)) and the increase in accuracy (Figure 3(a)) slow down. Subsequent addition of new rationale components has a diminished effect on loss reduction, making it difficult to identify the left rationale components.

This raises a question: is there an objective function that can indicate rationale, and does not involve either the mutual information or the sigmoid function of the neural network's output layer, thereby allowing for the possibility of avoiding diminishing marginal returns?

**Empirical observation on the norm of intermediate representation**. From the theoretical property mentioned in §3.2, we know that the representation's norm is an indicator for the degree of the network's utilization on an input. Besides, it does not involve either mutual information or the sigmoid function. We then empirically observe whether it faces the problem of diminishing marginal returns. Figure 3(c) shows how the norm varies with the proportion of true rationale components in the rationale candidate input within a trained standard RNP predictor. We see that it indeed does not face the problem of diminishing marginal returns. Instead, it grows even faster as the proportion of rationale components grows.

## 4.2 THE PRACTICAL METHOD

Unlike existing mainstream rationalization methods that add extra auxiliary modules, we do not change the model architecture of the vanilla RNP, thus preserving its conciseness and flexibility. The only needed modification is to replace the extractor's cross-entropy loss with the norm loss.

Compared to MMI-based methods (Equation 3), we remove the cross-entropy loss from the extractor's parameters $\theta_e$ and replace it with the norm:

$$\textbf{Extractor: } \min_{\theta_e} -log(\|Enc(Z)\|_2) \tag{8}$$

$$\textbf{Predictor: } \min_{\theta_p} H_c(Y, f_P(Z)|Z) \tag{9}$$

$$s.t., \ Z = f_E(X), \ (X,Y) \sim \mathcal{D}. \tag{10}$$

The predictor is trained to do the classification, and the extractor is trained to identify the rationales. During training, (9) and (8) are alternated. The practical implementation with Pytorch is in Appendix A.3. We call this method N2R (norm to rationale). Note that when we say MMI (see Equation (2)), it refers only to the objective of the extractor , and does not include the predictor. No matter how the extractor's objective changes, the predictor is always trained with the cross-entropy as it needs to do the classification.

To further show the potential and scalability of our N2R, we also verify the possibility of combining MMI and N2R (corresponding to the supplementary experiments in Figure 4). This combination is inspired by the empirical results of Figure 3. In the initial stage of training, the extractor has not yet identified enough true rationale components, leading to small gradients provided by the norm objective (Figure 3(c)), which is not efficient enough to guide the extractor in finding the rationale. However, at this stage, the MMI objective provides larger gradients (Figure 3(b)). In the later stages of training, the situation is reversed. Overall, the MMI and norm objectives complement each other, resulting in improved performance. We maintain the objective of the vanilla RNP (Equation 3) and

Table 1: Results on datasets from the BeerAdvocate Benchmark. We report the average results of five random seeds. Values in "()" are the standard deviations. Inter_RAT: Yue et al. (2023). NIR: Storek et al. (2023). A2I: Liu et al. (2024c).

| Datasets | Beer-Appearance | | | | | Beer-Aroma | | | | |
|---|---|---|---|---|---|---|---|---|---|---|
| Methods | S | Acc | P | R | F1 | S | Acc | P | R | F1 |
| RNP | 14.7 (0.7) | 78.2 (3.3) | 75.0 (0.5) | 59.7 (3.1) | 66.5 (2.1) | 15.2 (1.0) | 81.7 (2.4) | 67.0 (12.1) | 64.7 (8.8) | 65.8 (10.4) |
| Inter_RAT | 15.2 (1.1) | N/A | 57.0 (5.3) | 46.9 (2.3) | 51.4 (3.2) | 16.1 (0.7) | N/A | 57.9 (2.4) | 60.3 (2.5) | 59.0 (2.1) |
| NIR | 14.8 (0.4) | 78.2 (2.2) | 74.0 (1.3) | 59.0 (2.4) | 65.6 (2.0) | 15.4 (0.4) | 82.2 (3.2) | 65.4 (7.1) | 64.7 (6.2) | 65.1 (6.6) |
| A2I | 14.9 (0.3) | 81.0 (1.2) | 75.2 (0.9) | 60.6 (1.7) | 67.1 (1.3) | 14.8 (0.1) | 82.7 (2.3) | 69.4 (2.5) | 65.9 (2.6) | 67.6 (2.5) |
| N2R (ours) | 14.8 (0.5) | **82.3** (1.8) | **81.9** (2.7) | **65.3** (2.2) | **72.7** (2.1) | 14.9 (0.4) | **86.9** (4.5) | **70.2** (1.5) | **67.2** (1.3) | **68.7** (1.1) |

add the norm to the extractor's objective:

$$\textbf{Extractor: } \min_{\theta_e}[H_c(Y, f_P(Z)|Z) - log(\|Enc(Z)\|_2)] \tag{11}$$

$$\textbf{Predictor: } \min_{\theta_p} H_c(Y, f_P(Z)|Z) \tag{12}$$

$$s.t., \ Z = f_E(X), \ (X, Y) \sim \mathcal{D}. \tag{13}$$

We call this method MMI+N2R. The practical implementation with Pytorch is in Appendix A.4.

## 5 EXPERIMENTS

### 5.1 SETTINGS

**Baselines**. The main baseline for direct comparison is the vanilla MMI-based rationalization framework RNP (Lei et al., 2016), as RNP and our N2R match in selection granularity, optimization algorithm, and model architecture, which helps us to focus on our claims rather than some potentially unknown mechanisms. To show the competitiveness of our method, we also include several recently published methods that improve MMI with various regularizers: Inter_RAT (Yue et al., 2023), NIR (Storek et al., 2023), CR (Zhang et al., 2023), and A2I (Liu et al., 2024c), all of which have been discussed in §2.

**Datasets**. Although the rationale extraction process is unsupervised, the rationalization task requires comparing the rationale quality extracted by different models. This necessitates that the test set includes ground-truth rationales, which imposes special requirements on the datasets. Following the conventional setup in the field of rationalization, we employ four text classification datasets from two widely used benchmarks. Apart from the text data, we also include a graph classification dataset.

The text classification datasets are **Beer-Appearance, Beer-Aroma** (collected from the BeerAdvocate benchmark (McAuley et al., 2012)), **Hotel-Service, Hotel-Cleanliness** (collected from the HotelReviews benchmark (Wang et al., 2010)). We also use a graph classification dataset, called **BA2Motifs** (Ying et al., 2019), to verify generalizability. All of these datasets contain human-annotated ground-truth rationales on the test set, making it convenient to compare different methods' performance fairly. More details are in Appendix A.1.

**Implementation details**. Both the extractor and the predictor are composed of an encoder (RNN/Transformer/GCN) and a linear layer. We use three types of encoders: GRUs (following Inter_RAT and A2I, table 1 and 2), bert-base-uncased (following CR, table 3), and GCN (for the BA2Motifs dataset). For NIR and our N2R, considering they are both variants of the standard RNP, we first manually tune the hyperparameters for RNP, and then apply the hyperparameters to both NIR and N2R. For Inter_RAT, since it has originally been implemented on the beer-related datasets, we apply its original hyperparameters but only adjust the sparsity regularizer in Equation (5). For CR, we just keep the major settings ("bert-base-uncased", the Beer-Appearance dataset, and the sprasity of $10\%$) the same as it and copy its results from its original paper. We report the average results of five random seeds. More details are in Appendix A.2.

**Metrics**. Following the previous research of Inter_RAT and A2I, we mainly focus on the rationale quality, which is measured by the overlap between the human-annotated rationales and the model-selected tokens. The terms $P, R, F1$ denote precision, recall, and $F1$ score respectively. These metrics are the most frequently used in rationalization. The term $S$ represents the average sparsity of

Table 2: Results on datasets from the HotelReviews Benchmark.

| Datasets | Hotel-Service | | | | | Hotel-Cleanliness | | | | |
|---|---|---|---|---|---|---|---|---|---|---|
| Methods | S | Acc | P | R | F1 | S | Acc | P | R | F1 |
| RNP | 15.3 (0.3) | 96.5 (1.5) | 41.0 (1.5) | 54.6 (1.1) | 46.8 (1.4) | 15.3 (0.2) | 97.2 (1.6) | 28.1 (0.7) | 48.7 (1.1) | 35.6 (0.8) |
| Inter_RAT | 15.0 (0.8) | N/A | 28.9 (1.1) | 38.1 (1.9) | 32.8 (1.1) | 14.4 (1.1) | N/A | 27.2 (2.1) | 44.1 (2.4) | 33.6 (2.1) |
| NIR | 15.0 (0.3) | 96.9 (0.4) | 40.9 (1.5) | 53.5 (1.2) | 46.3 (1.4) | 15.5 (0.4) | 96.7 (0.9) | 28.0 (0.6) | 49.2 (1.0) | 35.7 (0.6) |
| A2I | 15.1 (0.4) | 96.7 (0.6) | 41.4 (1.7) | 54.6 (1.3) | 47.1 (1.5) | 15.3 (0.3) | 96.8 (1.0) | 28.8 (0.6) | 49.7 (1.1) | 36.5 (0.7) |
| N2R (ours) | 15.1 (0.2) | **97.4** (0.5) | **42.8** (0.5) | **56.3** (1.0) | **48.6** (0.6) | 14.8 (0.2) | **97.4** (0.4) | **31.8** (0.5) | **53.4** (0.8) | **39.8** (0.6) |

Table 3: Results with BERT encoder. " $*$ ": the results of baselines are obtained from the paper of CR (Zhang et al., 2023).

| Datasets | Beer-Appearance | | | | | Beer-Aroma | | | | |
|---|---|---|---|---|---|---|---|---|---|---|
| Methods | S | Acc | P | R | F1 | S | Acc | P | R | F1 |
| RNP* | 10.0 (n/a) | 91.5 (1.7) | 40.0 (1.4) | 20.3 (1.9) | 25.2 (1.7) | 10.0 (n/a) | 84.0 (2.1) | 49.1 (3.2) | 28.7 (2.2) | 32.0 (2.5) |
| A2R* | 10.0 (n/a) | 91.5 (2.2) | 55.0 (0.8) | 25.8 (1.6) | 34.3 (1.4) | 10.0 (n/a) | 85.5 (1.9) | 61.3 (2.8) | 34.8 (3.1) | 41.2 (3.3) |
| INVRAT* | 10.0 (n/a) | 91.0 (3.1) | 56.4 (2.5) | 27.3 (1.2) | 36.7 (2.1) | 10.0 (n/a) | 90.0 (3.0) | 49.6 (3.1) | 27.5 (1.9) | 33.2 (2.6) |
| CR* | 10.0 (n/a) | 92.4 (1.7) | 59.7 (1.9) | 31.6 (1.6) | 39.0 (1.5) | 10.0 (n/a) | 86.5 (2.1) | 68.0 (2.9) | 42.0 (3.0) | 49.1 (2.8) |
| N2R (ours) | 10.8 (0.3) | **93.5** (1.8) | **79.7** (4.1) | **36.3** (1.8) | **49.9** (2.5) | 10.0 (0.1) | **91.0** (3.6) | **74.3** (5.8) | **47.0** (3.7) | **57.6** (4.5) |

the selected rationales, that is, the percentage of selected tokens in relation to the full text. Since the sparsity of ground-truth rationales on these datasets is around $10\% \sim 20\%$, we adjust $s$ in Equation (5) to make $S$ be about $15\%$ (since Equation (5) is only a soft constraint, it cannot strictly limit $S$ to be exactly $15\%$.). $Acc$ stands for the predictive accuracy.

## 5.2 RESULTS

**Results on standard benchmarks**. Tables 1 and 2 show the results on the four text classification datasets. In terms of the rationale quality (F1 score), our N2R significantly outperforms the standard MMI-based method (i.e., RNP) and also beats its improved variants. Compared to the second-best results of previous methods, the relevant improvements of our N2R on these four datasets are $8.3\%$ (= $\frac{72.7-67.1}{67.1}$), $1.6\%$ (= $\frac{68.7-67.6}{68.8}$), $5.1\%$ (= $\frac{49.5-47.1}{47.1}$), and $9.0\%$ (= $\frac{39.8-36.5}{36.5}$), showing the competitiveness of replacing the MMI-based objective with our norm objective.

We also compare with a representative LLM, llama-3.1-8b-instruct, in Table 6 of Appendix A.7, and find that our simple N2R gets comparable results to it and can sometimes even outperform it.

**Results with BERT encoder**. We also follow CR to conduct experiments with pretrained bert-base-uncased as a supplement. Since some methods become highly sensitive to hyperparameters after switching to an over-parameterized BERT model (also supported by Remark 6.1 in (Zhang et al., 2023)), and our computational resources are insufficient for extensive hyperparameter tuning for these methods, we primarily compare our approach with methods that have already been implemented using BERT. The results are shown in Table 3. Our N2R still outperforms previous MMI-based methods significantly.

**Results with GCN encoder**. To show generalizability of our method, we expand the RNP framework to graph neural networks to conduct a supplement experiment. Since Inter_RAT, NIR, and CR are methods specifically designed for text data and are not suitable for graph tasks, we only compare our N2R with the standard RNP on the BA2Motifs dataset to show its effectiveness rather than competitiveness. For this dataset, we select a set of nodes for each graph as the rationale. The results are shown in Table 4. We see that our method is still effective when applied to graph neural networks. Note that our method is very simple and has the potential to be combined with more advanced methods in the future. However, since the interpretability of graph neural networks is not the focus of this paper, we leave it for future work.

**N2R can be further improved when combined with MMI**. To further verify the scalability of our N2R, we implement a variant of it by combining N2R and MMI criterion together, which is introduced at the end of §4. The comparison between vanilla MMI, N2R, and N2R+MMI on the datasets from the BeerAdvocate benchmark and HotelReviews benchmark are shown in Figure 4(a) and Figure 4(b), respectively.

Table 4: Results with GCN encoder on BA2Motifs.

| Methods | S | Acc | P | R | F1 |
|---------|---|-----|---|---|-----|
| RNP | 20.3 (2.5) | 95.2 (1.9) | 36.5 (5.5) | 36.5 (2.2) | 36.4 (3.8) |
| N2R | 20.1 (1.2) | 96.0 (1.9) | **40.1** (2.1) | **40.4** (4.4) | **40.2** (3.2) |

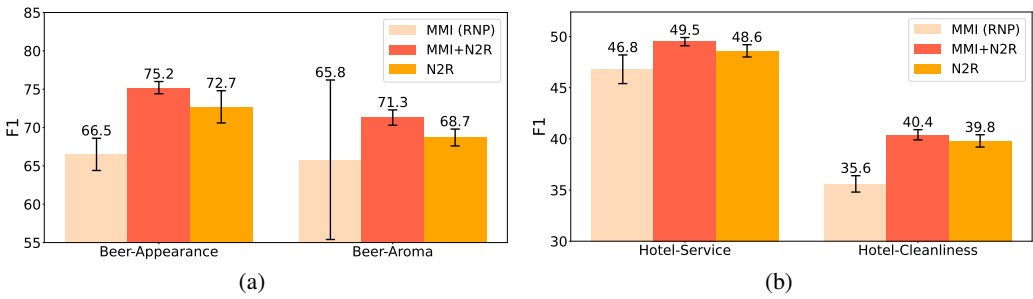

Figure 4: The comparison between vanilla MMI, N2R, and N2R+MMI on the datasets from (a) BeerAdvocate benchmark and (b) HotelReviews benchmark.

We can observe that, N2R on its own already significantly outperforms MMI, yet when combined with MMI, its performance improves even further. As compared to the vanilla MMI, the the relevant improvements of MMI+N2R on these four datasets are $13.1\%(= \frac{75.2-66.5}{66.5})$, $8.4\%(= \frac{71.3-65.8}{65.8})$, $5.8\%(= \frac{49.5-46.8}{46.8})$, and $13.5\%(= \frac{40.4-35.6}{35.6})$, respectively. This improvement of MMI+N2R aligns with the empirical results shown in Figure 3. In the initial stage of training, the extractor has not yet identified enough true rationale components, leading to small gradients provided by the OOD objective (Figure 3(c)), which is insufficient to effectively guide the extractor in finding the rationale. However, at this stage, the MMI objective provides larger gradients (Figure 3(b)). In the later stages of training, the situation is reversed. Overall, the MMI and OOD objectives complement each other, resulting in enhanced performance.

## 6 CONCLUSION AND FUTURE WORK

In this paper, we first analyze the diminishing marginal return limitation of the fundamental MMI-based objective in the XAI literature. Then, we propose to use the norm of the intermediate representation of rationale candidates to replace the MMI objective, which is inspired by OOD detection techniques. Our OOD-inspired objective not only outperforms the vanilla MMI, but also beats several recent variants. What's more, it can be easily combined with MMI, further validating its scalability and potential. Our work represents a pioneering attempt to bridge the fields of OOD detection and interpretability. This could potentially inspire researchers in the OOD field to adapt more OOD detection techniques to the XAI domain, thereby further advancing the development of XAI.

Most existing methods find explanations by reconstructing the model's final output, whereas we focus on identifying which parts of the input are utilized by the network during forward propagation. This is a new direction that can free explanation algorithms from relying on the model's final output. This property could have broader implications, such as potentially being used in the future to explain task-agnostic (i.e., not fine-tuned) pretrained encoders.

## 7 ACKNOWLEDGMENT

This work is supported by the National Key Research and Development Program of China under grant 2024YFC3307900; the National Natural Science Foundation of China under grants 62376103, 62302184, 62436003 and 62206102; Major Science and Technology Project of Hubei Province under grant 2024BAA008; Hubei Science and Technology Talent Service Project under grant 2024DJC078; and Ant Group through CCF-Ant Research Fund. The computation is completed in the HPC Platform of Huazhong University of Science and Technology

We sincerely thank the reviewers and the editor for their valuable feedback and efforts during the review process. They have helped a lot in improving the quality of this paper. We thank Lao Gao and Yuankai Zhang for providing the results on LLMs and image data.

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

Table 5: Statistics of datasets used in this paper

| Datasets | | Train | | Dev | | Annotation | | |
|---|---|---|---|---|---|---|---|---|
| | | Pos | Neg | Pos | Neg | Pos | Neg | S |
| Beer | Appearance | 16891 | 16891 | 6628 | 2103 | 923 | 13 | 18.5 |
| | Aroma | 15169 | 15169 | 6579 | 2218 | 848 | 29 | 15.6 |
| Hotel | Service | 50742 | 50742 | 6344 | 6344 | 101 | 99 | 11.5 |
| | Cleanliness | 75049 | 75049 | 9382 | 9382 | 99 | 101 | 8.9 |

## A  MORE DETAILS

### A.1  DATASETS

The statistics of the datasets are in Table 5. $Pos$ and $Neg$ denote the number of positive and negative examples in each set. $S$ denotes the average percentage of tokens in human-annotated rationales to the whole texts.

### A.2  EXPERIMENTAL DETAILS

The code and detailed running instructions will be made publicly available on GitHub after the paper is accepted. The code is now in an anonymous repository: `https://anonymous.4open.science/r/N2R-0E5E`. The anonymous repository will be closed in Nov 13, as we do not want people other than the reviewers to get the code.

The maximum sequence length is set to 256. We use the Adam optimizer Kingma & Ba (2015) with its default parameters, except for the learning rate (the learning rate is 0.0001). The temperature for gumbel-softmax is the default value 1. We implement the code with Pytorch on a RTX4090 GPU. We report the average results of five random seeds, and the seeds are [1,2,3,4,5].

For NIR and our N2R, considering they are both variants of the standard RNP, we first manually tune the hyperparameters for RNP, and then apply the hyperparameters to both NIR and N2R. For all datasets, we use a learning rate of 0.0001. The batchsize is 128 for the beer-related datasets and 256 for the hotel-related datasets. These hyperparameters are found by manually tune the standard RNP and are applied to both NIR and our N2R.

The core idea of NIR is to inject noise into the selected rationales. We use RNP as its backbone. A unique hyperparameter of NIR is the proportion of noise. Following the method in its original paper, we searched within $[0.1, 0.2, 0.3]$ and found that $0.1$ yielded the best results on most datasets, hence we adopted $0.1$ for it.

We found that the training of Inter_RAT is very unstable. To avoid potential unfair factors, our main settings are determined with reference to it. Except for the part about sparsity, we used its original hyperparameters for it.

For A2I, we contact its authors and get the code and hyperparameters of it.

For CR, we just keep the major settings ("bert-base-uncased", the Beer-Appearance dataset, and the sprasity of $10\%$) the same as it and copy its results from its original paper.

### A.3  IMPLEMENTATION DETAILS OF N2R

For a batch of $(X, Y)$, we first send $X$ to the extractor and get the rationale $Z$:
$$Z = f_E(X). \tag{14}$$
Then, we get a copy of $Z$ with the pytorch function "torch.detach()":
$$Z' = \text{torch.detach}(Z), \tag{15}$$
such that the following computation does not involve the extractor's gradients. Then, we send the rationale $Z'$ to the predictor and get the prediction $\hat{Y}$:
$$\hat{Y} = f_P(Z'). \tag{16}$$

Table 6: The quality of rationales extracted by llama-3.1-8b-instruct.

| Datasets | Beer-Appearance | | | | Beer-Aroma | | | |
|---|---|---|---|---|---|---|---|---|
| Methods | S | P | R | F1 | S | P | R | F1 |
| N2R (ours) | 14.8 (0.5) | 81.9 (2.7) | 65.3 (2.2) | **72.7** (2.1) | 14.9 (0.4) | 70.2 (1.5) | 67.2 (1.3) | **68.7** (1.1) |
| llama-3.1-8b (finetune) | n/a | 86.3 (n/a) | 46.2 (n/a) | 60.2(n/a) | n/a | 73.2 (n/a) | 50.6 (n/a) | 59.8 (n/a) |
| llama-3.1-8b (2 shot) | n/a | 15.4 (n/a) | 16.0 (n/a) | 15.7 (n/a) | n/a | 17.9 (n/a) | 24.2 (n/a) | 20.6 (n/a) |

| Datasets | Hotel-Service | | | | Hotel-Cleanliness | | | |
|---|---|---|---|---|---|---|---|---|
| Methods | S | P | R | F1 | S | P | R | F1 |
| N2R (ours) | 15.1 (0.2) | 42.8 (0.5) | 56.3 (1.0) | 48.6 (0.6) | 14.8 (0.2) | 31.8 (0.5) | 53.4 (0.8) | 39.8 (0.6) |
| llama-3.1-8b (finetune) | n/a | 77.3 (n/a) | 40.6 (n/a) | **53.3** (n/a) | n/a | 54.9 (n/a) | 31.3 (n/a) | 39.9 (n/a) |
| llama-3.1-8b (2 shot) | n/a | 45.3 (n/a) | 51.7 (n/a) | 48.3 (n/a) | n/a | 39.3 (n/a) | 43.0 (n/a) | **41.1** (n/a) |

Then, we update the predictor with the cross-entropy loss.

$$\min_{\theta_p} H_c(Y, \hat{Y}) \tag{17}$$

Note that this updating process with cross-entropy will not influence the extractor, since we have used "torch.detach()" for $Z$.

Then, we fix the parameters of the predictor, and only update the extractor. We first get the rationale candidate $Z$ with

$$Z = f_E(X). \tag{18}$$

And we then send it to the predictor's encoder to get $\|Enc(Z)\|_2$. Then, we update the extractor with

$$\min_{\theta_e} -log(\|Enc(Z)\|_2). \tag{19}$$

Then, we get into the next round to update the extractor and the predictor again.

## A.4 Implementation details of N2R+MMI

The implementation details are similar to those of Appendix A.3. The only difference is that we no more use "torch.detach".

## A.5 The minimum cross-entropy is equal to the entropy

The cross-entropy consists of two parts:

$$H_c(Y, \hat{Y}|Z) = H(Y|Z) + D_{KL}(P(Y|Z)\|P(\hat{Y}|Z)). \tag{20}$$

When we minimizing the cross-entropy $H_c(Y, \hat{Y}|Z)$ by adjusting the predictor's parameters, we are in fact minimizing $D_{KL}(P(Y|Z)\|P(\hat{Y}|Z))$. And we know that if the predictor is trained ideally, we have $\min D_{KL}(P(Y|Z)\|P(\hat{Y}|Z)) = 0$. Then, we have

$$\min_{\theta_p} H_c(Y, \hat{Y}|Z) = H(Y|Z). \tag{21}$$

## A.6 The detailed discussion about Equation (6)

In any cases, we have

$$I(Y; R_1, R_2) = I(Y; R_1) + I(Y; R_2|R1). \tag{22}$$
$$I(Y; R_1, R_2) - I(Y; R_1) - I(Y; R_2) = I(Y; R_2|R1) - I(Y; R_2). \tag{23}$$

The magnitude relationship between $I(Y; R_2|R1)$ and $I(Y; R_2)$ is arbitrary. In other words, there exists some scenarios where $I(Y; R_2|R1) < I(Y; R_2)$. In such cases, we have

$$I(Y; R_1, R_2) < I(Y; R_1) + I(Y; R_2). \tag{24}$$

In these cases, the mutual information faces the diminishing marginal returns problem.

## A.7 The rationales extracted by llama-3.1-8b-instruct

To further show the potential impact of rationalization in the era of LLMs, here we present the results of the experiments conducted with the llama-3.1-8b-instruct model. We perform both 2-shot prompting and supervised fine-tuning.

For 2-shot prompting, we provide the model with a negative text with its corresponding rationale, and a positive text with its corresponding rationale. For supervised fine-tuning, the supervison label is the classification label, since we perform unsupervised rationale extraction. We use 4*RTX 4090 24GB GPUs and LoRA to fine tune the models. We provide a detailed document in our anonymous code repository (https://anonymous.4open.science/r/N2R-0E5E/details_of_llms.pdf) to include all the details (including the prompt templates, LoRA fine-tuning parameter settings, and more).

In most cases, the model can output the rationale in the correct format. Figure 5 shows an example. But in 2-shot prompting, the model sometimes outputs additional parts along with the rationale (through manual observation, this situation does not occur frequently.). Figure 6 is another example. In such cases, we use gpt-3.5-turbo to extract the content within the quotation marks.

The results are shown in Table 6. LLMs are not good at counting, so we did not constrain the percentage length (i.e., sparsity) of the rationale extracted by the model. Compared to our N2R, llama-3.1 does not have a crushing advantage. On two out of four datasets, our N2R outperforms llama-3.1. And on the left two datasets, N2R achieves comparable results to llama-3.1. Besides, our N2R can be applied to graph data, while it is not easy to do so for LLMs.

> **Task:** Sentiment classification about Beer's appearance
> **Input:** Pours a rather crisp yellow almost orange with a thin head. The aroma is dominated by sweet malts with just a slight hoppiness dancing in the background. The taste does have a surprising amount of hoppiness for a Pilsner. There is a good maltiness to it as well, but citrus hops just slightly overpower. The beer is very light and refreshing. This makes for an excellent summer session beer.
> **Expected output:** 1|pours a rather crisp yellow almost orange with a thin head .
> **llama-3.1 output:** 1|pours a rather crisp yellow almost orange

Figure 5: An example of llama's output. Here "1" means that the class label $Y$ is positive. And the words after "|" represent the rationale.

> **Task:** Sentiment classification about Beer's appearance
> **Input:** 22oz bottle pouted into a Goblet: Opaque orange with a light, white, creamy head that was not all that well retained but full of carbonation, but did settle into a small thin cap. The aroma was more Belgian triple than IPA, Sweet and malty. The taste is a very nice balance of the two styles. A little more hops, but balanced very nice with the sweetness of the malt and fruit. The beer had a medium to full body, perhaps a little too thick for my taste, but still good. The beer had a nice bitter dry aftertaste and was well carbonated. The beer was fairly easy to drink give the ABV, but after the 22oz, I was pretty well done. Overall, a good beer and probably the first one of the side projects that I think the brewery should consider brewing on a regular basis.
> **Expected output:** 1|opaque orange with a light , white , creamy head that was not all that well retained but full of carbonation , but did settle into a small thin cap .
> **llama-3.1 output:** positive|The overall tone of the review is positive, with phrases such as "a very nice balance of the two styles", "nice bitter dry aftertaste", "well carbonated", and "overall, a good beer" indicating a favorable opinion of the beer.
> **GPT-3.5-restructured output:** 1|a very nice balance of the two styles nice bitter dry aftertaste well carbonated overall, a good beer

Figure 6: An example of llama fails to output the rationale in the right format.

## A.8 MORE RESULTS OF DIMINISHING MARGINAL RETURNS

Figure 7 shows the results on more datasets.

## A.9 A VISUALIZED EXAMPLE OF EXTRACTED RATIONALES

Figure 8 shows a visualized example of rationales extracted by different methods on the Beer-Aroma dataset.

Figrue 9 shows a visualized example of rationales extracted by different methods on the CUB (see Appendix A.10) dataset.

## A.10 THE RESULTS ON CUB DATASET

**TL;DR**. The results are in Table 7.

The CUB dataset (Wah et al., 2011) contains photographs of birds annotated by species, totaling around 11.7k samples. In our experiment, we follow Sagawa et al. (2019) to categorize them into

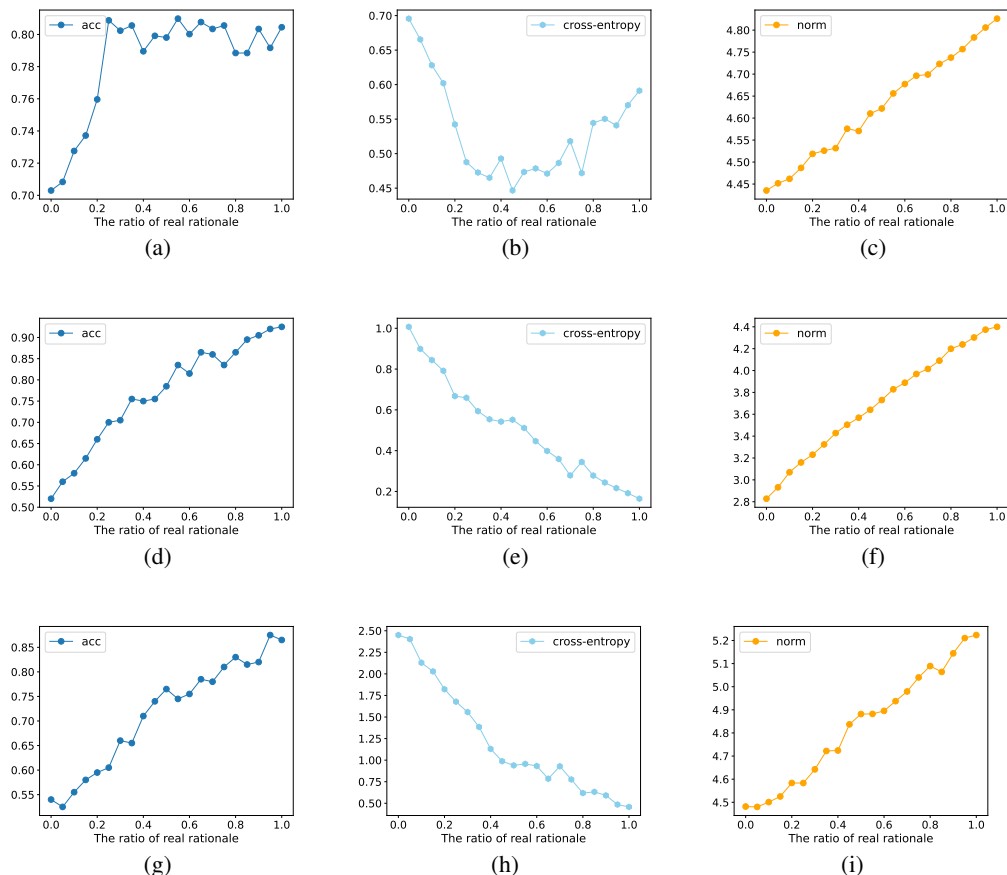

Figure 7: The prediction accuracy, cross-entropy loss, and the norm of the representation through the neural network vary with the proportion of true rationale components in the rationale candidate input within a trained standard RNP predictor. (a)(b)(c): the Beer-Appearance dataset. (d)(e)(f): the Hotel-Service dataset. (g)(h)(i): the Hotel-Cleanliness dataset.

two categories: waterbird (albatross, auklet, cormorant, frigatebird, fulmar, gull, jaeger, kittiwake, pelican, puffin, tern, gadwall, grebe, mallard, merganser, guillemot, or Pacific loon) and landbird (ani, blackbird, bobolink, bunting, cardinal, catbird, Yellow Breasted Chat, Eastern Towhee, Chuck Will Widow, cowbird, Brown Creeper, crow, cuckoo, finch, Northern Flicker, flycatcher, goldfinch, grackle, grosbeak, hummingbird). Sagawa et al. (2019) annotated each image with the bird's silhouette in order to study the overfitting problem, which can serve as the ground-truth rationale.

The classification task on this dataset is challenging due to severe overfitting problems. Since other baseline rationalization methods are specifically designed for text data and are not suitable for image data, we only compare with vanilla MMI (i.e., RNP) to validate our method's effectiveness rather than competitiveness. We also compare with a vanilla classifier (ResNet18) to verify the effectiveness of our method in improving classification accuracy.

**Details and metrics**. For this image classification dataset, the extractor and predictor in Figure 1 are different from those used in performing text tasks. The goal of the extractor is to select a portion of the pixels from an image as the rationale, which can be seen as performing a binary classification on each pixel, similar to image segmentation. So, we use an U-Net to be the extractor. And for the predictor, we use a ResNet18. The meaning of F1 is the same as the one used in Table 1 (the overlap between the extractor-selected pixels and ground truth pixels). We add an additional metric, IoU (a common metric for image segmentation), to further measure the rationale quality.

**Label** (Beer-Aroma): Positive.
**Prediction:** Positive.
**Input:** medium brown , fairly clear and light penetrable . small thin ring of white foam . sweet and rich malt smells . very nutty ( roasted almond , pecan ) with some caramel . rich almond and toffee notes , some bitter cocoa or slight espresso maybe . not as sweet as the nose implies , some dryness and alcohol present . [unknown] alcoholic . fairly strong carbonation , medium heavy mouthfeel . pretty drinkable .

(a) RNP

**Label** (Beer-Aroma): Positive.
**Prediction:** Positive.
**Input:** medium brown , fairly clear and light penetrable . small thin ring of white foam . sweet and rich malt smells . very nutty ( roasted almond , pecan ) with some caramel . rich almond and toffee notes , some bitter cocoa or slight espresso maybe . not as sweet as the nose implies , some dryness and alcohol present . [unknown] alcoholic . fairly strong carbonation , medium heavy mouthfeel . pretty drinkable .

(b) Inter_RAT

**Label** (Beer-Aroma): Positive.
**Prediction:** Positive.
**Input:** medium brown , fairly clear and light penetrable . small thin ring of white foam . sweet and rich malt smells . very nutty ( roasted almond , pecan ) with some caramel . rich almond and toffee notes , some bitter cocoa or slight espresso maybe . not as sweet as the nose implies , some dryness and alcohol present . [unknown] alcoholic . fairly strong carbonation , medium heavy mouthfeel . pretty drinkable .

(c) NIR

**Label** (Beer-Aroma): Positive.
**Prediction:** Positive.
**Input:** medium brown , fairly clear and light penetrable . small thin ring of white foam . sweet and rich malt smells . very nutty ( roasted almond , pecan ) with some caramel . rich almond and toffee notes , some bitter cocoa or slight espresso maybe . not as sweet as the nose implies , some dryness and alcohol present . [unknown] alcoholic . fairly strong carbonation , medium heavy mouthfeel . pretty drinkable .

(d) A2I

**Label** (Beer-Aroma): Positive.
**Prediction:** Positive.
**Input:** medium brown , fairly clear and light penetrable . small thin ring of white foam . sweet and rich malt smells . very nutty ( roasted almond , pecan ) with some caramel . rich almond and toffee notes , some bitter cocoa or slight espresso maybe . not as sweet as the nose implies , some dryness and alcohol present . [unknown] alcoholic . fairly strong carbonation , medium heavy mouthfeel . pretty drinkable .

(e) N2R (ours)

Figure 8: A Visualized example of rationales on the Beer-Appearance dataset. The underlined words are human-annotated ground-truth rationales. Model-selected rationales are highlighted with different colors.

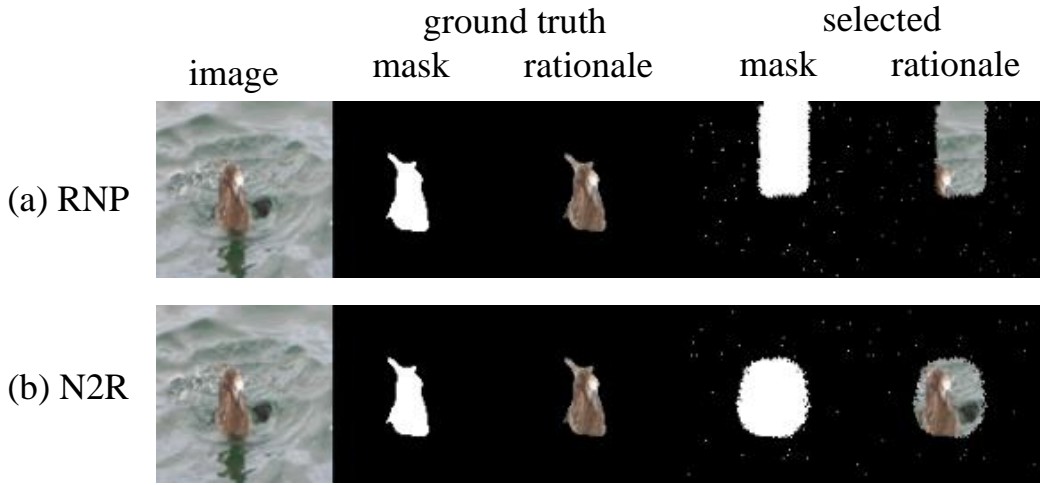

Figure 9: A Visualized example of rationales on the CUB dataset.

**Results**. The results are shown in Table 7. On this challenging image classification dataset, our N2R still outperforms the MMI-based method RNP. As for the improvement in classification accuracy compared to the vanilla classifier ResNet18, one possible reason is that our N2R retains information relevant to classification while removing irrelevant noise (however, the poor performance of RNP may be due to its failure to find comprehensive useful information). This phenomenon is consistent with the findings of Wu et al. (2022), which suggest that extracting rationales can enhance generalizability to some extent.

We also provide a visualized example for this dataset in Figure 9. We find that RNP selects only a small part of the bird but a large part of the water (shortcut) to classify this image as "waterbird". While the N2R-selected rationale includes the whole bird.

Table 7: The results on the CUB dataset.

| Methods | S | Acc | P | R | F1 | IoU |
|---|---|---|---|---|---|---|
| RNP | 15.5 (1.1) | 81.3 (0.7) | 33.3 (2.1) | 37.0 (4.7) | 35.0 (3.2) | 21.3 (2.4) |
| N2R (ours) | 15.8 (1.1) | **86.5** (0.6) | **46.1** (5.4) | **52.5** (4.8) | **49.1** (4.9) | **32.6** (4.2) |
| Classifier (ResNet18) | n/a | 85.0 (0.8) | n/a | n/a | n/a | n/a |

Table 8: The results on the MovieReview dataset. "*": The results of the baselines are copied from Inter_RAT(Yue et al., 2023) (they did not report the classification accuracy).

| Methods | S | Acc | P | R | F1 |
|---------|---|-----|---|---|-----|
| RNP* | 20.0 | - | 35.6 | 21.1 | 24.1 |
| A2R* | 20.0 | - | 48.7 | 31.9 | 34.9 |
| INVRAT* | 20.0 | - | 33.9 | 24.3 | 28.3 |
| Inter-RAT* | 20.0 | - | 35.7 | 35.8 | 35.7 |
| N2R | 20.3 (2.2) | **88.4** (2.1) | **45.6** (2.1) | 31.7 (1.6) | **37.4** (1.9) |
| Classifier | n/a | 86.4 (2.3) | n/a | n/a | n/a |

Table 9: Results with BERT encoder. The dataset is the most widely used Beer-Appearance. "∗": results obtained from the paper of CR (Zhang et al., 2023).

| Datasets / Methods | Beer-Appearance | | | | | Beer-Aroma | | | | |
|--------------------|-----------------|---|---|---|---|------------|---|---|---|---|
| | S | Acc | P | R | F1 | S | Acc | P | R | F1 |
| RNP* | 10.0 (n/a) | 91.5 (1.7) | 40.0 (1.4) | 20.3 (1.9) | 25.2 (1.7) | 10.0 (n/a) | 84.0 (2.1) | 49.1 (3.2) | 28.7 (2.2) | 32.0 (2.5) |
| A2R* | 10.0 (n/a) | 91.5 (2.2) | 55.0 (0.8) | 25.8 (1.6) | 34.3 (1.4) | 10.0 (n/a) | 85.5 (1.9) | 61.3 (2.8) | 34.8 (3.1) | 41.2 (3.3) |
| INVRAT* | 10.0 (n/a) | 91.0 (3.1) | 56.4 (2.5) | 27.3 (1.2) | 36.7 (2.1) | 10.0 (n/a) | 90.0 (3.0) | 49.6 (3.1) | 27.5 (1.9) | 33.2 (2.6) |
| CR* | 10.0 (n/a) | 92.4 (1.7) | 59.7 (1.9) | 31.6 (1.6) | 39.0 (1.5) | 10.0 (n/a) | 86.5 (2.1) | 68.0 (2.9) | 42.0 (3.0) | 49.1 (2.8) |
| N2R (ours) | 10.8 (0.3) | **93.5** (1.8) | **79.7** (4.1) | **36.3** (1.8) | **49.9** (2.5) | 10.0 (0.1) | 91.0 (3.6) | **74.3** (5.8) | **47.0** (3.7) | **57.6** (4.5) |
| Classifier | n/a | 93.0 (2.6) | n/a | n/a | n/a | n/a | **91.6** (3.1) | n/a | n/a | n/a |

## A.11 RESULTS ON MOVIEREVIEW DATASET

**TL;DR**. The results are in Table 8.

MovieReview (Pang & Lee, 2004) is a text classification dataset with much longer texts (the average length is 774 words) as compared to the datasets used in Table 1 and 2. Recently, DeYoung et al. (2020) annotated rationales for this dataset so that it can be used for the rationalization task.

We follow the settings of Inter_RAT (Yue et al., 2023) to conduct experiments on this dataset. The encoders for extractor and the predictor are both GRUs. And the word embedding is GloVe-100d. The maximum sentence length is set to be 1024.

We compare with the baselines already implemented by Inter_RAT and copy the results from it. We also compare with a vanilla classifier (GloVe+GRU) to see the classification performance.

The results are shown in Table 8. We see that our N2R still outperforms MMI-based methods on this long-text challenging dataset.

## A.12 COMPARISON WITH BERT CLASSIFIER.

We compare the classification accuracy with a vanilla classifier implemented with BERT on the Beer-Appearance and Beer-Aroma datasets. The results are shown in Table 9. Our N2R gets even better accuracy than the BERT classifier. One possible reason is that our N2R retains information relevant to classification while removing irrelevant noise. This phenomenon is consistent with the findings of Wu et al. (2022), which suggest that extracting rationales can enhance generalizability to some extent.

## A.13 THE CONVERGENCE SPEED

Figure 10 shows a comparison of convergence speed between RNP and N2R. The batchsize is 128 and the learning rate is 0.0001. We see that N2R converges much faster than RNP, which means that we can train N2R with fewer steps and thus saving the computational costs.

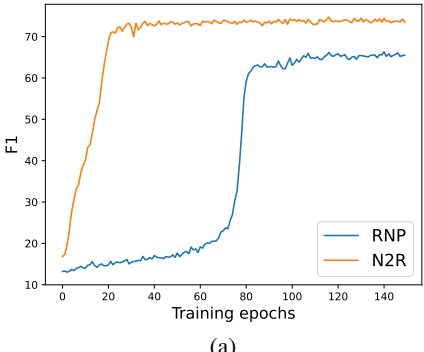 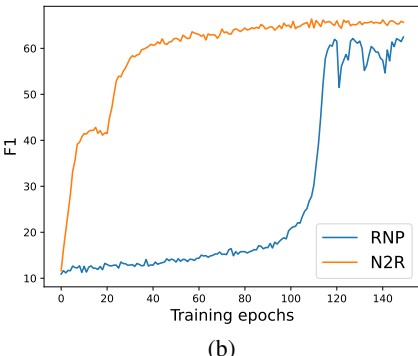

(a)                 (b)

Figure 10: The comparison of convergence speed between RNP and N2R. (a) The Beer-Appearance dataset. (b) The Beer-Aroma dataset.

### A.14 THEORETICAL SUPPORT FOR USING NORMS TO DISTINGUISH BETWEEN RATIONALES

**Lemma 1 ((Cai et al., 2013))** *Let $U$ and $V$ be two random points on a $p$-dimensional unit hypersphere $\mathbb{R}^p$, and $O$ is the origin. Let $\Theta$ be the angle between vector $\overrightarrow{OU}$ and vector $\overrightarrow{OV}$, then*

$$\Pr\left(|\Theta - \frac{\pi}{2}| \geq \epsilon\right) \leq K\sqrt{p}(\cos\epsilon)^{p-2}, \tag{25}$$

*for all $p \geq 2$ and $\epsilon \in (0, \pi/2)$, where $K$ is a universal constant.*

Lemma 1 is the Proposition 5 of (Cai et al., 2013). And it tells us that "all high-dimensional random vectors are almost always nearly orthogonal to each other" (Cai et al., 2013). As the dimension $p$ increases, $\Theta$ gradually converges to $\pi/2$.

If two vectors are orthogonal to each other, then their dot product will be zero.

We consider a simple case. Consider a FFN layer without the bias term. And its weight matrix is $W \in \mathbb{R}^{m \times n}$, and the input is $X \in \mathbb{R}^m$. The output of the FFN layer is $XW \in \mathbb{R}^n$. Each dimension in $XW$ is the dot product of $X$ and a column vector of $W$. The column vectors of $W$ represent some directions the model can handle. For a well-trained FFN, the directions are determined by its training data and represent the information of it learns.

We use the lower case $x$ to denote a specific input.

Consider an input $x_1$ does not contain any information that has been learned by $W$. For example, $W$ is part of a classifier trained to distinguish between cell phones and water cups, and $x_1$ is a photo of a grassy. Although $x_1$ is not truly random noise, since $W$ has never been trained to recognize grassland, $x_1$ acts like random noise to $W$. So it is likely that $x_1$ is orthogonal to those column vectors in $W$. So, every dimension of $x_1 W$ will approach zero and thus $\|x_1 W\|_2$ will also approach zero.

If an input $x_2$ is full of the information learned by $W$, then it probably matches the directions of $W$. And thus $\|x_2 W\|_2$ will be high.

Then, what would be the intermediate state of $x_1$ and $x_2$? **We use $V \in \mathbb{R}^m$ to denote an arbitrary column vector in $W$.** We assume an input $x_3$. The first $k$ dimensions of $x_3$ (denoted as $x_3^{1:k}$) contain useful information, and the last $n - k$ dimensions represent uninformative noise. We have that

$$x_3 \cdot V = x_3^{1:k} \cdot V^{1:k} + x_3^{k+1:n} \cdot V^{k+1:n} \tag{26}$$

$x_3^{k+1:n}$ consists of uninformative noise, so we have $x_3^{k+1:n} \cdot V^{k+1:n} \approx 0$. Thus $x_3 \cdot V = x_3^{1:k} \cdot V^{1:k}$.

We assume that the useful information is uniformly distributed in each dimension of $X$ and $V$. For an input $x$, we denote $x^i$ as the $i$-th dimension of it. Formally:

**Assumption 1** *If an input $x$ contains $k$-dimensional useful information (for simplicity, we always assume that the useful information is in the first $k$ dimensions), we assume that the informativeness between different dimensions is the same:*

$$x^i \cdot V^i = x^j \cdot V^j, \ \forall i \leq k, j \leq k. \tag{27}$$

With this assumption in place, we can quantitatively analyze why our method is not affected by the problem of diminishing marginal returns.

Consider that the proportion of the gold rationale components in an extractor-selected rationale candidate $x_a$ is $\frac{k_a}{n}$ (i.e., there are $k_a$ of $n$ dimensions in $x_a$ represent useful information). When we add more gold rationale components to it to make the proportion be $k_a'$ and the new rationale candidate be $x_a'$. We will have that

$$\frac{\|x_a W\|_2}{\|x_a' W\|_2} = \frac{k_a}{k_a'}. \tag{28}$$

**Proof**.

$$\begin{aligned}
x_a W &= [x_a \cdot V_1, \cdots, x_a \cdot V_n] \\
&= [x_a^{1:k_a} \cdot V_1^{1:k_a}, \cdots, x_a^{1:k_a} \cdot V_n^{1:k_a}] \\
&= k_a [x_a^1 \cdot V_1^1, \cdots, x_a^1 \cdot V_n^1],
\end{aligned} \tag{29}$$

where the third equation is from Assumption 1.

Similarly, we have

$$x_a' W = k_a' [x_a'^1 \cdot V_1^1, \cdots, x_a'^1 \cdot V_n^1], \tag{30}$$

$x_a'$ is got from replacing the uninformative part of $x_a$ (i.e., the last a few dimensions) with useful information, so we have $x_a'^1 = x_a^1$. Thus we have $\frac{x_a W}{x_a' W} = \frac{k_a}{k_a'}$ and $\frac{\|x_a W\|_2}{\|x_a' W\|_2} = \frac{k_a}{k_a'}$.

**Conclusion**. Ideally, the norm metric we designed should increase approximately linearly as the proportion of gold rationale in the rationale candidates grows, thus avoiding the problem of diminishing marginal returns like MMI-based methods.

Although we made an assumption that may not hold in reality, it was made to facilitate a better quantitative analysis, and the conclusions drawn at least have a qualitative trend. The trends in Figure 7(c)(f)(i) also verify this trend.

### A.15 A TOY EXAMPLE FOR BETTER INTUITIVE UNDERSTANDING OF OUR METHOD

Consider a network consists of only a linear layer (without bias) and has a low-rank weight matrix $M$ that, after elementary row transformations, takes the following form:

$$\begin{bmatrix} 1 & 1 & 1 \\ 0 & 1 & 1 \\ 0 & 0 & 1 \\ 0 & 0 & 0 \end{bmatrix}$$

$A$ and $B$ represent inputs with and without the model's learned information respectively. After undergoing corresponding transformations (existing research indicates that after performing certain elementary transformations on the weight matrix, a corresponding transformation can be found for the inputs, resulting in identical outputs (Ainsworth et al., 2023)), they are likely to take the forms like: $A = [1, 0, 0, 0]$, $B = [0, 0, 0, 1]$. The informative input $A$ usually matches more with the non-zero parts of $M$. And $\|AM\|_2 = \sqrt{3} > \|BM\|_2 = 0$. (The column vectors of the weight matrix represent certain directions in high-dimensional space (determined by the learned information), but random noise $B$ generally does not fall along these directions as it does not contain the corresponding information.)

