# OpenReview forum: "Breaking Free from MMI: A New Frontier in Rationalization by Probing Input Utilization"
_ICLR.cc/2025/Conference — ICLR 2025 Poster_

### Official Review · Reviewer_3eT9 · 2024-10-30

**Soundness:** 3
**Presentation:** 4
**Contribution:** 2
**Rating:** 6
**Confidence:** 5

**Summary:**

The authors propose a novel approach for extracting rationales in a select-predict-like pipeline, replacing the traditional MMI loss criterion  with a new objective that is independent of the output prediction.

They claim that MMI suffers from diminishing marginal returns, that is, the model can select a portion of the input as rationales and still predict the true label with insignificantly lower accuracy.

Therefore, instead of relying on the output prediction for evaluating the rationales extraction, they prob the weight matrix of neural networks norm to detect snippets that the model truly utilize.

**Strengths:**

- Model agnostic method (They propose a new metric for rationale extraction which is regardless of the model itself as long as you have access to the weights).
- Their method falls in the self-explanation category which guarantees faithful rationales.
- Potential to be extended to image classification and graph neural network.
- Mixing N2R and MMI improves the performance even better as MMI works better in the early stages of finding rationales and N2R works better when the models already detected few.

**Weaknesses:**

- Works like UNIREX already discussed the problem you mentioned about MMI by introducing "sufficiency" and "comprehensiveness" metrics.
- The extractor and the predictor are not trained cooperatively (end-to-end) due to the non-differentiability of rationale space.
- You should put the task performance in the results.
- Visualization of some extracted rationale would be great.
- You didn't provide the hyperparameter (for example in Eq. 5) in your experiments.

**Questions:**

- I was wondering what are the metrics when you pass gold rationales to the predictor directly. This would indicate the maximum performance you could reach.
- I was wondering what is the performance of your framework w.r.t. OOD data. This helps to understand if the model learn to rely on plausible rationales instead of statistical shortcuts
- BeerAdvocate is more sentence level rationales, evaluating on token level rationales (like e-SNLI) which is more challenging would be interesting.

---

> ### Author Response · Authors · 2024-11-20
> **We sincerely thank you for dedicating your time and expertise to review our paper. Your insightful comments and suggestions are highly valued and appreciated.**
>
> **Weakness1 (UNIREX already discussed comprehensiveness)**. We agree that UNIREX has discussed comprehensiveness and its excellent work has inspired many subsequent studies, including ours.
>
> However, UNIREX and we contribute to the issue of "comprehensiveness" from different perspectives. UNIREX **identifies** (from the outcome perspective) that the rationales found by the model are not comprehensive enough, and has designed a comprehensiveness regularizer to fix this problem. However, the remaining untouched question is, why existing models fail to find comprehensive rationales. Our unique contribution is the **analysis** of the cause of this issue, specifically the diminishing marginal returns characteristic of MMI (as claimed in L112-115). Understanding the cause of this issue can help us find better solutions.
>
> In summary, UNIREX's contribution is the observation and identification of this important issue, while our contribution is the analysis of the cause of this issue.
>
> **Weakness2 (not trained end-to-end)**. We are sorry, but this is a misunderstanding. Our method was trained end-to-end. The non-differentiability problem was solved with gumbel-softmax (the same as A2I).
>
> **Weakness3 (task performance)**. Thank you for your suggestion; now the new Tables 1-4 all include task performance (i.e., Accuracy).
>
> **Weakness4 (Visualization examples)**. Thank you for your suggestion, we now add the examples in Appendix A.9.
>
> **Weakness5 (hyperparameters)**. In our initial version, we only provided the learning rate, batch size, and their search strategies (Appendix A.2), but forgot to mention the sparsity and continuity constraints in Eq. 5. This is because these two hyperparameters were directly inherited from the A2I code and were not modified. In Eq.5, $\lambda_1=11$, and $\lambda_2=12$.
>
>
> **Q1 (what are the metrics when you pass gold rationales to the predictor directly)**. We are sorry but we are not sure what you mean by "metrics". Could you please be more specific? The points on the far right of the x-axis in Figure 3 represent the results when the gold rationale is directly input into the RNP predictor.
>
> **Q2 (OOD data and statistical shortcuts)**. Thank you for your suggestion. We now add a dataset CUB, where OOD data is a challenging problem. The results and discussions are in Appendix A.10. Our N2R still outperforms MMI-based RNP in terms of both rationale quality and classification accuracy. From the visualized example in Fig.9, we find that RNP selects the background water (shortcut) as the rationale for "waterbird", while our N2R does not.
>
> **Q3 (evaluating on token level rationales )**. Although the rationales in BeerAdvocate are annotated at the sentence level, our extractor actually performs selection at the token level. Additionally, the hotel-related datasets we used have span-level rationales. Perhaps these clarifications can address your concerns to some extent.
>
> For e-SNLI, we do not find a sufficient number of suitable baselines in a short time. Therefore, we add another dataset with token-level rationales, MovieReview. The results are in Appendix A.11. N2R still outperforms recent MMI-based methods.

---

> > ### Comment · Reviewer_3eT9 · 2024-11-24
> > **Thank you for clarifications**
> >
> > Thank you for clarifying the doubt and making modifications to the draft.
> > By metric (regarding Q1), I meant performance. Sorry for the misunderstanding.
> > Overall, my questions are addressed.

---

> > > ### Author Response · Authors · 2024-11-24
> > > **Thank you very much**
> > >
> > > Thank you for your thoughtful reviews and suggestions. Best wishes to you and yours!

---

### Official Review · Reviewer_meUo · 2024-11-02

**Soundness:** 2
**Presentation:** 2
**Contribution:** 2
**Rating:** 6
**Confidence:** 3

**Summary:**

The study introduces an alternative approach to Maximum Mutual Information (MMI) for extracting rationales in explainable AI (XAI). Traditionally, MMI-based methods have been used to extract relevant features by maximizing information shared between input rationales and predictions. However, the authors argue that MMI suffers from diminishing marginal returns, limiting its efficacy in fully capturing critical rationale components. They propose using the norms of input representations, aligning extracted rationales with utilized parts of the network's weight matrix. This novel method Norm to Rationale (N2R), reportedly outperforms MMI-based methods across several datasets (text and graph-based). Authors use GRUs, BERT and GCN as an encoders, and several text classification and one graph classification dataset.

**Strengths:**

- The use of encoder output norms for rationale extraction is novel, and the experiments provide convincing motivation for using this approach over MMI-based methods.
- The theoretical analysis of MMI-based methods is clear and engaging.
- The proposed method is promising due to its simplicity and potential for generalizability.
- The authors conduct experiments with multiple random seeds and report results with standard deviations, which significantly improves the rigor of the work and reinforces confidence in the performance of the approach.

**Weaknesses:**

- The experiments are limited to datasets with annotated rationales, and the authors do not present results showing how their approach improves classification quality in more advanced real-world datasets.
- The experiments focus solely on text and graph domains. The claim of generalizability would be strengthened by including additional domains.
- The impact of norm-based rationale selection on computational resources, particularly for larger models, is not discussed. Additionally, the authors omit details and experiments regarding the properties of the encoder and extractor, such as whether the encoder should have a specific architecture or how scaling the encoder affects the method.
- While the empirical results are compelling, the theoretical foundation for norm-based rationale extraction could benefit from additional depth. More detailed mathematical insights into why norms better capture input utility and how the input interacts with the encoder's weight matrix could strengthen the approach.

**Questions:**

I find the approach interesting, but I have a few questions and suggestions that could further improve the study:

- How does this approach actually improve classification quality in real-world scenarios? The paper mentions MMI limitations, but do these truly impact final performance? I understand that classification performance isn’t the primary goal of explainable AI, but it would be interesting to see if there’s a noticeable effect.
- You claim generalizability across domains, yet the experiments are limited to text and graphs. Could this method be extended to other domains, such as speech or even image classification?
- It is unclear if there are specific limitations regarding encoder and extractor architectures. Could you clarify this? Additionally, it would be interesting to see how the approach scales with (a) an increasing number of encoder parameters and (b) longer input text lengths. I believe such experiments would enhance the work, making it more valuable for tasks like in-context learning and RAG prompt improvements.

Minor Remarks and Questions:

- It is unclear why you use the logarithm in Eq. (8) and subsequent equations. Is this related to Fig. 3c?
- Figure 3 is somewhat confusing: there are no y-labels, and the caption lacks references to panels (a), (b), and (c).
- Figure 2 appears after Figure 3; it may be better to reorder them for clarity.

---

> ### Author Response · Authors · 2024-11-20
> **Response part 1**
>
> We are truly grateful for your detailed review and the thoughtful suggestions you provided.
>
> **Weakness1&Q1 (classification quality in real-world scenarios)**. We are sorry, but we are not sure about the specific concern of you by mentioning "real-world scenarios". Could you please be more specific?
>
> Except for BA2Motif, all the other datasets we use are real-world datasets. And in the experiments of Table 1 and 2, our method indeed has a higher prediction accuracy compared to the MMI-based methods.
>
> Do you mean that we need to consider more challenging datasets? Now we add two challenging datasets, CUB and MovieReview, for image classification and text classification respectively. The CUB (image classification) dataset is considered challenging by previous studies due to its overfitting problem. And the Movie dataset is challenging due to its long texts. More details and results are in Appendix A.10 and A.11.
>
> Aside from rationalization methods, we also implement a vanilla non-interpretable classifier on the above two datasets. And our N2R gets even better classification results as compared to the vanilla classifier. Detailed discussion is in Appendix A.10.
>
> And for the datasets previously used in our submission, we also implement a vanilla classifier with BERT for the Beer-Appearance and Beer-Aroma datasets. The results are in Appendix A.11. We still get comparable results with BERT classifier.
>
> **Weakness2&Q2 (experiments on more domains aside from text and graph)**. Thank you for your suggestion. We now extend our method to the image domain. We implement the extractor with a U-net and the predictor with ResNet18. The extractor selects a set of pixels to be the rationale. We evaluate our method on a image classification dataset CUB. Our N2R is still effective on this domain. More results and discussions are in Appendix A.10.
>
> **Weakness3&Q3 (computational resources, encoder architectures, and scalability)**.
>
> **Computational resources**. A typical predictor consists of a feature encoder and an MLP head. The encoder first transforms the input into a feature vector, which is then fed into the MLP to produce the output. The only additional operation we perform is calculating the norm of the feature vector, which has a very minor computational cost, making the extra overhead during forward propagation negligible. The extra computation may be more apparent during backward propagation. In RNP, the cross-entropy backpropagation goes through both the predictor and extractor, thus we denote the computational load to be 2. N2R's cross-entropy loss only passes through the predictor, making this part of the computation a 1, while the norm loss first passes through the predictor and then the extractor, making its computation a 2, thus the total computation of N2R is 3. Therefore, the backward propagation cost of N2R is about 1.5 times that of RNP. However, although our method has higher costs than the original RNP, it remains lower than other more advanced methods, as they introduce additional auxiliary modules as regularization terms, with computational loads more than twice that of RNP.
>
> Besides, we find our N2R converges much faster than RNP, which means that we can train N2R with fewer steps (N2R needs less than half steps) and thus saving the computational costs. The comparison of convergence speed between RNP and N2R are in Appendix A.13.
>
> The memory usage (MB) for the GRU-based settings (Table 1) are as follows:
> | Bachsize| 128 | 256 | 512 |
> |---|---|---|---|
> | RNP | 1998 | 3316 | 5372 |
> | N2R | 1998 | 3318 | 5374 |
>
> The training time (on a RTX4090 GPU) for the Beer-Appearance dataset is
> | Bachsize | 128 | 256 | 512 |
> |---|---|---|---|
> | RNP (100 epochs) | 725s | 460s | 343s |
> | N2R (100 epochs) | 1093s | 584s | 394s |
> | N2R (50 epochs) | 547s | 292s | 197s |
>
> Since the backpropagation of N2R (Eq.8 and 9) is conducted in two steps, the memory usage does not increase compared to RNP. Given that N2R converges faster (Fig.10 shows that N2R requires less than half the training epochs of RNP), the actual training time is shorter than that of RNP. Moreover, because each step of our two-step backpropagation consumes less computational resources than RNP, the advantages of N2R become more pronounced when increases in computational load (such as increased batch size) make GPU power the bottleneck.

---

> ### Author Response · Authors · 2024-11-20
> **Part 2**
>
> **Encoder architectures**. A typical predictor consists of a feature encoder and an MLP head. The encoder first transforms the input into a feature vector, which is then fed into the MLP to produce the output. The only additional operation we perform is calculating the norm of the feature vector. Thus, our method is encoder-agnostic, as we do not directly manipulate the encoder, but instead operate on the output vectors of the encoder. There are also no specific limitations regarding the extractor architecture, as our extractor is the same as RNP.
>
> Our experiments are conducted on different encoder architectures, including RNN-based (GRUs), Transformer-based (BERT), and GNN-based (GCN). And we have now added a CNN-based (ResNet18) experiment (see Weakness2&Q2). We are not sure if we have correctly understood your concern; could you please specify a bit more?
>
> **Scalabilit (more model parameters and longer texts)**.
> Thank you for your valuable suggestions. We now add an additional MovieReview dataset which consists of longer texts. The results are in Appendix A.11. N2R still outperforms MMI-based methods on both rationale quality and classification accuracy.
>
> As for the impact of model size, our current experiments have covered small models (GRU and GCN), medium-sized models (ResNet18), and relatively large models (BERT).
> We observe that as the model size increases, the improvement of N2R over RNP also grows (the improvement is minor when using GCN and GRU, more significant in experiments using ResNet18, and even greater when using BERT). This may be due to larger models being more likely to correctly classify based on very subtle clues, which exacerbates the difficulty for MMI to identify complete clues. However, an increase in model size tends to make the distinction between informative rationale and uninformative noise more pronounced in terms of norm (the weight matrix can be viewed as a series of column vectors. Mathematically, given a k-dimensional vector, the probability that it is orthogonal to a randomly sampled k-dimensional noise vector increases with the dimension k).
>
> Could you please specify your concerns more with more details so that we can identify areas that need clarification or additional experiments?
>
> **Weakness4 (theoretical support)**. Thank you for your suggestion. We have now added theoretical support for our method in Appendix A.14, which covers two aspects.
>
> **First, why the norm can be used to distinguish between informative rationale and uninformative noise:** the overall idea is that in high-dimensional space, two random vectors are highly likely to be orthogonal, thus the norm of the noise input multiplied by the weight matrix approaches zero.
>
> **Second, why our method is not affected by the problem of diminishing marginal returns.** We have shown that, under certain ideal assumptions, the norm increases almost linearly with the increase in useful information in the input, which also matches the trend observed in Figure 7(c)(f)(i).
>
> **Minor issues**.
> 1. Using a logarithm is an empirical attempt, aimed at ensuring sufficient gradient in the initial phase of training (as the early norm growth in Fig.3 is not significant, adding a log can amplify the gradient), and preventing gradient explosion in the later stages of training. The logarithm is also a common trick in other fields.
> 2. We apologize for the confusion caused. The legend in the figure already indicates the meaning of the y-coordinates, so we omitted the y-label. We have now added references to (a), (b), and (c) in the caption.
> 3. We will revise it in the camera-ready version (for now, to avoid confusion in referencing during review and revision, we have not made changes on it).

---

> > ### Comment · Reviewer_meUo · 2024-11-25
> >
> > Thank you for your comprehensive and thoughtful response to my comments. I appreciate the effort you have put into addressing the points I raised.
> >
> > Regarding my earlier comment about "real-world scenarios," I thought that it could be better to try to apply the method to, for example, data filtering. I now realise that my suggestion was not entirely relevant in the context of your work. After reading your response and the updated manuscript, I believe the paper is complete and strong without such applications.
> >
> > I also want to commend you for extending your method to the image domain. This addition significantly enhances the paper's scope and demonstrates the versatility of your approach. The detailed experiments and results presented in the appendices further strengthen the contributions of your work.
> >
> > One minor suggestion for improvement: Appendix A.15 could benefit from some revisions to make it cleaner and easier to follow.
> >
> > Based on these improvements and clarifications, I am happy to raise my score. Thank you again for your diligent efforts and clear explanations.

---

> > > ### Author Response · Authors · 2024-11-26
> > > **Thank you very much**
> > >
> > > Thank you for your thoughtful reviews and encouragement. Best wishes to you and yours!

---

### Official Review · Reviewer_hZKo · 2024-11-04

**Soundness:** 3
**Presentation:** 3
**Contribution:** 2
**Rating:** 6
**Confidence:** 4

**Summary:**

This work suggests extracting crucial rationales from an input of language models using the weight matrices of these language models. The proposed method is an alternative to the more traditional MMI method. It should help us move towards more explainable AI by showing which parts of the input are crucial for making a decision.

**Strengths:**

- The work investigates an important topic: XAI.
- Authors repeat each experiment several times and report standard deviation.

**Weaknesses:**

It is very hard to understand the contribution:

- The method description in abstract, introduction and many other places of the paper seems to be simply wrong. Weight matrices of LLMs (excluding "attention weights"; but the paper doesn't tell anything about attention weights) are not dependent on the particular input, so they can not be used for estimating the parts of inputs. The method description in the paper should be rewritten completely.

**UPDATE**: The authors have addressed the previously mentioned weakness and clarified their intended meaning. They made a revision of the paper. It is now very clear from the revised text that they studied the interaction between the weights and the input representations from the previous layer of the model (see discussion below).

- I suppose that the authors could mean components of the embedding matrix, not the weight matrix? But this suggestion still leads to confusion: I looked into the code and it still wasn't clear how exactly the model's embeddings were used in calculations. E.g. the function "`get_embeddings`" from embeddings.py is not used in other scripts at all. At the same time, "`get_glove_embedding`" function was used; but there is not even a single mention of GloVe in the paper itself, so there is a contradiction again. I urge the authors to make their code more clear and consistent with the paper text.

**UPDATE**: The authors have clarified this inconsistency in their comments, and the revised paper now mentions GloVe.

- The reasoning about the rank of "weight matrices" also sounds unclear and not founded enough. It is again unclear, whether weight matrices or embedding matrices are considered; besides, in the context of neural networks, it is hardly possible to talk about "zero matrix rank" for sure in any case. In a strict sense, "zero rank" would mean that the whole matrix is exactly zero which is very rare case in practice. Probably authors mean some approximation of the matrix rank, but the exact formula for that approximation was never written down.

**UPDATE**: The authors provided somewhat more justified explanation for this part (see the discussion below and the revised paper).

- Figure 2 doesn't look correct, it's meaning is unclear (see "Questions" part). Its description also looks unclear and not founded enough.

Other issues:

- If we assume again that authors mean embeddings and not weight matrices (as follows from common sense and from "4.2 The practical method" part of the paper), it is still not clear, what will happen if the input isn't connected to the task at all. E.g. if the input is like "John went for a walk", and the task is to estimate, whether he liked the beer aroma or not. It seems that the suggested approach (or, at least, my, very vague understanding of it), isn't able to correctly handle such situations.

**UPDATE**: Figure 2 and OOD example were commented in the discussion.

Minor issues:

- Citations in lines 167-173 should be put into braces.

**UPDATE**: After a thoughtful discussion with the authors and reviewing the revised version, I now have a clearer understanding of the method, and it seems reasonable to me. As a result, I have increased my main score from 3 to 6. The other scores have been adjusted accordingly.

**Questions:**

- What does x axis of Figure 2 mean?
- What do notations in Equation 2 mean? (they are not defined in the text)

---

> ### Author Response · Authors · 2024-11-20
> **Thank you deeply for taking the time to thoroughly review our paper.**
>
> **Overall clairfication**.
> We are so sorry to make such a huge misunderstanding. But we do refer to  weight matrices (e.g., the green RNN/Transformer in Fig.1) and not to the embedding matrix. We do not mention the embedding matrix because it is just an untrainable dictionary and thus a part of the input itself (i.e., the embedding matrix is considered as a pre-processing tool of the $X$ in Fig.1 and $X$ can be considered as a sequence of dense word vectors rather than one-hot vectors).
>
> Consider that the rationale candidate $Z$ (Fig.1) is a sequence of word vectors (i.e., the output of a embedding matrix), after sending $Z$ to an encoder (e.g., RNN/Transformer), we get a sequence of word representations $B=RNN(Z)$. We then get a single vector $C$ by pooling: $C=maxpooling(B)$. Finally , the $||Enc(Z)||_2$ in our method (L354) is obtained by $||Enc(Z)||_2=||C||_2$.
>
> **W1 (weight matrices cannot be used for estimating the parts of inputs)**. We agree that weight matrices are not dependent on the particular input, but we have different oppinions about "they cannot be used for estimating the parts of inputs".
>
> What we mean isn't that the weight matrix itself can be used to estimate inputs, but rather the interaction between the weight matrix and the input is used to determine whether an input can be utilized. If the representation norm of a rationale candidate selected by the extractor is close to 0 after passing through the weight matrix, it indicates the input cannot be utilized. Conversely, if the rationale candidate can be effectively utilized, its representation will have a larger norm after passing through the weight matrix. **The reason for this property is Sec 3.2 and empirically verified by Fig 3(c).**
>
> Here is a toy example for better intuitive understanding.
> Consider a network consists of only a linear layer (without bias) and has a low-rank weight matrix $M$ that, after elementary row transformations, takes the following form:
> | 1 | 1 | 1 |
> |---|---|---|
> | 0 | 1 | 1 |
> | 0 | 0 | 1 |
> | 0 | 0 | 0 |
>
> $A$ and $B$ represent inputs with and without the model's learned information respectively. after undergoing corresponding transformations (existing research indicates that after performing certain elementary transformations on the weight matrix, a corresponding transformation can be found for the inputs, resulting in identical outputs [1]), they are likely to take the forms like: $A=[1,0,0,0]$, $B=[0,0,0,1]$. The informative input $A$ usually matches more with the non-zero parts of $M$. And $||AM||_2=\sqrt{3}>||BM||_2=0$. (The column vectors of the weight matrix represent certain directions in high-dimensional space (determined by the learned information), but random noise B generally does not fall along these directions as it does not contain the corresponding information.)
>
> That's how the weight matrice can be used for distinguishing between informative and uninformative rationale candidates. And Fig.3(c) and Fig.7(c)(f)(i) are the practical verifications of the above property.
>
>
> **W2 (GloVe)**. We used three different encoders (GRU/BERT/GCN) in Sec.5.
> GloVe is used in conjunction with the GRU encoder, whereas the other two encoders do not require an additional external embedding matrix. We apologize for not mentioning this. And the GRU+GloVe setup follows the baseline Inter_RAT (L420). The code is consistent with the paper, as our method is independent of the embedding matrix.
>
>
> **W3 (confusing term "zero-rank components")**. We are sorry to make such a misunderstanding. When we mention "zero rank components", we in fact refer to the subspace that corresponds to the last row of matrix $M$ in response **W1**--a row with all values being zero. However, $M$ is a transformed matrix; the corresponding positions in the original weight matrix are not necessarily zero, so it's not proper to refer to them as "positions with values of zero." Therefore, we use "zero-rank components" to denote these redundant rows.
>
>
> **W4 (Fig.2)**. Fig.2 is a qualitative toy example (instead of a quantitative practical experiment) of the Sigmoid function to help intuitively understand the diminishing marginal returns from the perspective of gradient saturation. It was cited in L289-292. The x-axis represents the proportion of genuine gold rationale components in a extractor-selected rationale candidate. The middle of the x-axis indicates the absence of gold rationale components, representing complete noise. The further away from the center, the higher the content of gold rationale components.
>
> **W5 (task-irrelevant inputs)**. Since "John went for a walk" does not contain the model's learned information, it is an OOD input that falls outside the space that the model can handle. Just like the dots in the lower left of Figure 3(c), these dots are the results of inputting sentiment-irrelevant text into the predictor (sentiment classifier).
>
> [1] Git Re-Basin: Merging Models modulo Permutation Symmetries. ICLR 2023.

---

> > ### Comment · Reviewer_hZKo · 2024-11-21
> > **Thank you for clarifications**
> >
> > **WEIGHTS AND EMBEDDINGS**
> >
> > Thank you for your very detailed response. It has helped me understand the source of the confusion regarding weight matrices and embeddings that we had.
> >
> > I now realize that we have a difference in how we’re using the term "embedding". In my review, I was using the word embedding in a broader sense, which includes any learned vector representations of input tokens, including intermediate representations that come from intermediate layers in the model, as well as final representation from the final layer. The examples of papers with broad usage of this term: NV-Embed: Improved Techniques for Training LLMs as Generalist Embedding Models by Lee et al; "An Interpretability Illusion for BERT" by Bolukbasi et al etc.
> >
> > E.g. in my terms, $Enc(Z)$ is also an embedding from the final layer of the network. In contrast, you use the term "embedding" in a narrower sense, referring to the pre-trained lookup table that maps words or tokens to continuous vector spaces. If you still want to strictly reserve the term "embedding" for an input embedding only, using "final layer representation" instead is also okay (as you already do it in some parts of the paper).
> >
> > > Here is a toy example for better intuitive understanding.
> >
> > Thank you for this toy example, it is easy to understand and makes your point much more clear. In your example, I would refer to $AM$ and $BM$ as an embeddings of $A$ and $B$ correspondingly and say that we make our conclusion by investigating the norm of these embeddings.
> >
> > In general, I would encourage you to incorporate some key points about weight matrices from your response into the introduction or method description in your paper, to make it more clear as well. The "toy example" could also be made the part of Appendix.
> >
> > ---
> >
> > **MATRIX RANKS**
> >
> > The reasoning about matrix ranks still look very strange to me.
> >
> > > W3 (confusing term "zero-rank components"). We are sorry to make such a misunderstanding. When we mention "zero rank components", we in fact refer to the subspace that corresponds to the last row of matrix M in response W1 - a row with all values being zero.
> >
> > I think it is misleading to refer to the zero row of the matrix as "zero rank component", and the matrix itself clearly doesn't have zero rank if it has such row. I think the paper would benefit from re-formulating your thoughts about these components.
> >
> > Whiel re-reading some parts of the paper, I also noticed that you cite two well-known works: "Intrinsic dimensionality explains the effectiveness of language model fine-tuning" and "Lora: Low-rank adaptation of large language models" as a support for your point that "Neural networks usually have low-rank weight matrices" in line 093 and other parts of your paper. But these works don't support this point. Roughly speaking, these works say that you can optimize only low-dimensional subspace of the weights of neural network to solve downstream task. They don't claim anything about the rank of weight matrices of the model itself, after or before finetuning. It is clear that weight matrices of the model itself may have full rank, be changed with update matrix of a low rank, and still retain full rank.

---

> ### Author Response · Authors · 2024-11-21
> **Thank you very much for your patient response and detailed valuable suggestions. They have been truly helpful to us.**
>
> **1. (using "final layer representation")** Thank you very much for your valuable suggestion. We have now revised the introduction and Fig.1 accordingly. And we have added the toy example to Appendix A.16.
>
> **2. (matrix ranks)**
>
> **About the references in L93**.
>  Although [1] did not directly mention that the weight matrix is low-rank, they noted that the original large number of parameters can be derived from a smaller number of parameters through linear projection (please refer to the screenshot in Appendix A.15). This indicates that many parameters in the original weight matrix are linear combinations of other parameters. When the parameters of some rows (or columns) can all be derived from combinations of other rows (or columns), the matrix is considered low-rank.
>
>  As for the second paper Lora, we are grateful for your reminder. we realize that Lora indeed mentioned that the update matrix is low-rank, not that the original matrix is low-rank. We have now changed the supporting literature in L93 to the same as L99 (i.e., [2]).
>
>   Apart from the subtle implication in [1], our Section 3.2 is mainly borrowed from [2]. And Lemma D.4 in [2] shows theoretically that neural networks tend to learn low-rank weight matrices (see Fig. 12).
>
>  The details of the above discussion are in Appendix A.15.
>
>  We greatly appreciate your reminder; to avoid causing any confusion, we have changed the support reference to [2].
>
>  **About the term "zero rank component"**. We know it is a confusing term, and we are really grateful for your suggestion. We have now completely removed this term, and accordingly revised the statements in the abstract (L18-25), introduction (L93-98), and Sec 3.2 (L242-249).
>
>  Overall, for $XW$, we consider each column vector in the weight matrix $W$ as a direction, and the linear combinations of these column vectors occupy a subspace of the hypersphere (e.g., a low-dimensional manifold). If the input falls within this subspace (i.e., the input vector's direction matches the directions of those column vectors), the model will be able to utilize it. If an input is orthogonal to these directions (according to Lemma 1 in Appendix A.14, for a given vector and a randomly sampled noise vector, they are likely to be orthogonal if the dimension is large enough), then the model cannot utilize this input. We refer to this subspace as the model's "capability subspace", and use the terms 'outside/inside the capability subspace' to replace the previous terms 'zero/non-zero rank components'.
>
>  [1] Intrinsic dimensionality explains the effectiveness of language model fine-tuning.
>  [2] Deep neural networks tend to extrapolate predictably.
>
> Thank you again for your thoughtful suggestions.

---

> > ### Comment · Reviewer_hZKo · 2024-11-25
> > **Raising the score**
> >
> > Thank you for your clarifications. Unfortunately, I don’t have enough time to thoroughly check all the math in the new appendices. However, I have re-read the main sections of the revised paper, along with the shorter appendices, and read your discussion with the other reviewers. I now agree with Reviewer f8iZ and your abstract that the method is simple (at least, when described properly), but that simplicity does not detract from its value.
> >
> > After further reflection on the experimental section, I conclude that while the number of datasets is small and they appear relatively simple, they are more or less reasonable for the scope of the study.
> >
> > Overall, I believe the paper deserves a higher score than I initially assigned (as reflected in the updates).
> >
> > I am sorry for my initial low score and very harsh review, which were due to a lack of patience at the time.
> >
> > ---
> >
> > Last but not least, I would suggest updating the Abstract in the OpenReview system to ensure it matches the one in the latest version of your paper.

---

> > > ### Author Response · Authors · 2024-11-26
> > > **Thank you very much**
> > >
> > > Thank you very much for your thoughtful reviews and patient suggestions, which have helped us a great deal. Best wishes to you and yours!

---

### Official Review · Reviewer_f8iZ · 2024-11-04

**Soundness:** 2
**Presentation:** 2
**Contribution:** 2
**Rating:** 5
**Confidence:** 3

**Summary:**

+ The paper first identifies an issue of current MMI methods: once part of the rationales are detected, it is hard to find the remaining parts due to diminishing marginal returns.

+ Targeting this issue, the paper empirically analyzes the norm of the rationale's representation in the training process, considering the relationship between the information richness and the representation norm.
+ Based on the analysis results, the paper proposes to use/add the log representation norm to the Extractor's training loss, and the proposed method achieves better performance compared to several baselines.

**Strengths:**

+ The identified issue of current MMI methods is meaningful and insightful.
+ The proposed method makes sense intuitively. It is also a simple way to tackle the identified issue and seems to be effective.

**Weaknesses:**

+ Missing some key evaluation results
  + What's the accuracy of the Predictor (i.e., task accuracy) if taking as input the rationale extracted by BERT-based methods and llama-3.1-8b-instruct?
  + Why did you evaluate BERT-based methods on ONLY Beer-Appearance? Is there any specific reason for not showing results on the other three datasets?
+ It would be better to study what the improvement in the numbers actually means. Humans may have very different annotations when asked to extract a rationale. For instance, in the toy example (line 266), I personally prefer taking only R2 as the rationale as it is quite enough to make the prediction (and thus I can give a more concise rationale than R1+R2). Therefore, it is unclear whether the number improvement really means a better rationale (e.g., a rationale that avoids missing information that is actually essential). One possible way is to do some case study or human analysis to compare rationales extracted by different methods.

**Questions:**

+ Comparing Table 1 and Table 3, I found that BERT-based methods (i.e., taking BERT as the encoder) seem to perform worse than GRU-based methods on Beer-Appearance. It is not intuitive for me as BERT is expected to be a stronger model. Do you have any explanations?

---

> ### Author Response · Authors · 2024-11-20
> **Thank you for taking the time to carefully review our work and provide constructive feedback.**
>
> **Weakness 1-1 (about accuracy)**.
> We have now added the accuracy for BERT experiments (Table 3). Since our main focus was on the rationale quality rather than prediction accuracy, we may have overlooked this metric initially when filling out the forms. Thank you very much for your reminder.
>
> The reason we did not report the prediction accuracy for llama is because we found that there is a significant likelihood that responses from Llama would not contain explicit "positive/negative" classifications (for instance, in the hotel-service dataset, 50.5% of responses from the llama-finetune model did not specify a category, and this figure was 53.0% for hotel-cleanliness dataset). If we were to consider these responses without clear categories as incorrect predictions, the accuracy would appear very low and could be misleading. Considering that our main comparison is on the rationale quality and we do not claim our classification ability is superior to LLMs, hence we did not report this prediction accuracy to avoid unnecessary confusion.
>
> **Weakness 1-2 (BERT experiments on other datasets)**.
> We have now added experiments on the other dataset of the BeerAdvocate benchmark, Beer-Aroma (Table 3). And our N2R still beats all the baselines. Since CR did not conduct experiments on the other two Hotel benchmark datasets, and due to the lack of comparative baselines, we did not perform experiments on these two datasets.
>
> While using BERT, we found that some baseline methods were very sensitive to hyperparameters. Since previous baselines tend to design additional regularization terms, this adds complexity to the hyperparameter search process. Due to the limitations of our computational resources, we could not perform sufficient hyperparameter searches for these methods (as noted in L457-460). To avoid unfair comparisons arising from hyperparameter issues, we primarily compare against other baselines that have already been reproduced by the authors of CR as supplementary to the main experiments, verifying the generalizability of our model with different encoders. Since experiments involving BERT may be affected by unknown factors such as hyperparameter tuning, making it unsuitable for directly validating our claim, we did not choose it as the main experiment.
>
> The reason we conducted experiments using BERT on only one dataset is that the BERT experiment serves merely as a supplementary experiment to the main study (L457-459) and is intended to verify the generalizability of our model when different encoders are used. Therefore, we selected the most commonly used Beer-Appearance dataset (which has been used by nearly all methods in the field) as a representative. Our main experiments (Tables 1 and 2) follow the settings of A2I and Inter_RAT.
>
>
> **Weakness 2 (It would be better to study what the improvement in the numbers actually means)**.
> Thank you for your suggestion, we have now add a case study in Appendix A.9.
>
> Here is the explanation for your intuition "I personally prefer taking only R2 as the rationale as it is quite enough to make the prediction". For your convenience, we copy the mentioned toy example here: "⋯⋯ The food is very delicious (=R1), and I like it very much. (=R2) ⋯⋯".
>
> We agree R2 is enough to make the prediction, and this is exactly why MMI-based methods cannot identify R1+R2. But R2 lacks comprehensiveness. Although R2 itself can indicate the sentiment label, it does not tell where the sentiment tendency comes from. For example, R2 cannot tell me why I ike the food. Is it because the food tastes good or smells good, or just has an attractive packaging?
>
> The comprehensiveness is important in some high-risk decisions. For example, in AI-assisted diagnosis, if a person is infected with both Virus A and Virus B, regardless of whether Virus A or Virus B is chosen as the explanation, the model can successfully classify the person as sick. However, providing only one virus as the explanation could mislead the model user (a doctor) in their subsequent decision-making.
>
> **Question1 (BERT is expected to be a stronger model)**.
>
> There are several possible reasons. First, Table 3 follows the CR setting by setting the rationale sparsity to 10%, compared to 15% in Table 1, which might be a more challenging setup. Second, as mentioned in L457-459, using BERT is more sensitive to hyperparameter choices, which may not be perfect (thus we mainly compare with baselines already implemented with BERT, and do not treat these comparisons as the main experiment, but only as supplementary to the main experiment).
>
> Third, BERT's capabilities might be too strong, enabling it to classify based on very subtle clues. For MMI-based methods, MMI can only find a very small part of the correct rationale. Our method is not affected by this factor (as it does not seek rationales by maximizing prediction accuracy), hence our method's relative improvement is even more pronounced under settings using BERT.

---

> > ### Author Response · Authors · 2024-12-02
> >
> > Dear Reviewer f8iZ,
> >
> > As the ICLR discussion deadline is approaching, I wanted to kindly check if you have any further questions or need any clarifications regarding the paper.
> >
> > Thank you for your time and effort in reviewing our submission.

---

### Meta-Review · Area_Chair_QdbV · 2024-12-24

**Metareview:**

This paper points out an important weakness due to the serial nature of rationalization with MMI: “once part of the rationales are detected, it is hard to find the remaining parts due to diminishing marginal returns.” The work provides an alternative by probing input utilization via the norm of the rationale’s training time representation and the weights of the layer. The method changes the training of the rationale extractor, and presents comparisons with different related methods.

**Strengths:** The method is clearly motivated and well presented. Reviewers found the empirical results satisfactory. The method seems generalizable to multiple domains.

**Weaknesses:** Some of the methods used in the paper are somewhat older (BERT, GloVe) raising questions about how the method fares under more improved models.

**Reason for acceptance (poster)** The paper points out the diminishing marginal returns issue with a very standard interpretability method: MMI and presents a novel approach for rationalization that addresses this issue, therefore potentially leading to further work in this space.

**Additional Comments On Reviewer Discussion:**

Based on the reviewers’ suggestions, the authors have added a human analysis and experiments for image classification, as well as an efficiency analysis. The reviewers also very patiently engaged with the authors and helped each other’s understanding of the work.

---

### Decision · Program_Chairs · 2025-01-22

Accept (Poster)